# *Arabidopsis* FHY3 and FAR1 integrate light and strigolactone signaling to regulate branching

Yurong Xie [1], Yang Liu[1], Mengdi Ma[1], Qin Zhou[2], Yongping Zhao[2], Binbin Zhao[2], Baobao Wang[1], Hongbin Wei[3] & Haiyang Wang [3,4✉]

Branching/tillering is an important parameter of plant architecture and is tightly regulated by both internal factors (such as plant hormones) and external factors (such as light conditions). How the various signaling pathways converge to coordinately regulate branching is not well understood. Here, we report that in *Arabidopsis*, FHY3 and FAR1, two homologous transcription factors essential for phytochrome A-mediated light signaling, and SMXL6/SMXL7/SMXL8, three key repressors of the strigolactone (SL) signaling pathway, directly interact with SPL9 and SPL15 and suppress their transcriptional activation of *BRC1*, a key repressor of branching, thus promoting branching. In addition, FHY3 and FAR1 also directly up-regulate the expression of *SMXL6* and *SMXL7* to promote branching. Simulated shade treatment reduces the accumulation of FHY3 protein, leading to increased expression of *BRC1* and reduced branching. Our results establish an integrated model of light and SL coordinately regulating *BRC1* expression and branching through converging at the *BRC1* promoter.

[1] Biotechnology Research Institute, Chinese Academy of Agricultural Sciences, 100081 Beijing, China. [2] Graduate School of Chinese Academy of Agricultural Sciences, 100081 Beijing, China. [3] State Key Laboratory for Conservation and Utilization of Subtropical Agro-Bioresources, South China Agricultural University, 510642 Guangzhou, China. [4] Guangdong Laboratory for Lingnan Modern Agriculture, 510642 Guangzhou, China. ✉email: whyang@scau.edu.cn

Branching (tillering in cereal crops) is a major component of plant architecture and critical determinant of crop productivity. A critical stage of branch development is bud outgrowth after the formation of axillary buds, and this process is intricately regulated by endogenous developmental cues and various environmental signals[1]. Recent molecular genetic studies have revealed a conserved *TB1/FC1/BRC1* pathway repressing axillary bud outgrowth in both monocots and dicots. *TEOSINTE BRANCHED 1* (*TB1*), which encodes a TCP-family transcription factor, was initially identified as a key repressor of lateral branching in maize[2]. Later studies showed that loss-of-function mutants of the genes homologous to *TB1* in rice and *Arabidopsis* (*FINE CULM1* or *FC1* in rice, *Branched1* or *BRC1* in *Arabidopsis*) display a similar excess tillering/branching phenotype[3–5]. Moreover, recent studies showed that strigolactone (SL), a newly identified plant hormone, represses bud outgrowth in both monocots and dicots[6–8]. Further, it was shown that SLs repress tiller outgrowth in rice through promoting the degradation of a central repressor protein, D53[9,10]. Similarly, it has also been shown that in *Arabidopsis*, SL-induced degradation of three D53-like SUPPRESSOR OF MORE AXILLARY GROWTH2-LIKE proteins SMXLs (SMXL6, SMXL7, and SMXL8) leads to activation of *BRC1* to inhibit branching[11].

Substantial evidence suggests that the MIR156-SPL (*SQUA-MOSA PROMOTER BINDING PROTEIN-LIKE*) module plays an important role in regulating diverse aspects of plant growth and development, ranging from vegetative to reproductive phase transition, branching, leaf development, flowering time, panicle/tassel architecture, fruit ripening, fertility, lateral root development, and abiotic stress responses[12,13]. *SPLs* encode a family of plant-specific transcription factors that share a highly conserved DNA-binding domain, the SBP domain. It has been shown that *Arabidopsis* SPL9 and SPL15 negatively regulate branching by suppressing axillary meristem initiation and that their loss-of-function mutants display an enhanced branching phenotype[14,15]. Similarly, a single-nucleotide polymorphism that escapes miR156 targeting or increase *OsSPL14* expression via epigenetic regulation confer an ideal plant architecture to rice, including fewer unproductive tillers, stronger culm, enlarged panicle and ultimately, enhanced grain yield. Thus OsSPL14 was also named *Ideal Plant Architecture1* (*IPA1*) or *WEALTHY FARMER'S PANICLE* (*WFP*)[16,17] (*IPA1* was used hereafter). Further analysis revealed that IPA1 could directly bind to the promoter of *OsTB1* (*FC1*), to regulate branching/tillering[18]. Recent studies further showed that IPA1 can directly bind to the promoter of *D53* and that D53 protein can inhibit the transcriptional activation activity of IPA1 by direct physical interaction, thus forming a feedback loop to repress the expression of *D53*[19]. Such a mechanism has also been shown to be conserved in wheat[20]. These results clearly indicate that the SL signaling pathway and the MIR156/SPL regulatory module converge at the promoter of *TB1/FC1/BRC1* to coordinately regulate branching in plants.

Light is a major environmental factor that regulates branching pattern in plants. When plants sense a reduction in red to far-red light ratio (R/FR) due to competition for light from the neighbor plants (vegetation proximity) or grown under canopy shade, they initiate a set of adaptive responses collectively termed shade avoidance syndrome (SAS), including rapid shoot elongation, more erect leaves, accelerated leaf senescence, early flowering, reduced branching, and ultimately decreased biomass and seed yield if the shading is persistent[21,22]. It has been shown that under simulated shade conditions (low R/FR ratios), the expression of *BRC1* is elevated, leading to suppression of branching activity[23]. We previously showed that under simulated shade conditions, accumulation of the phytochrome-interacting factors (PIFs) proteins increases, and they directly bind to the promoters of

several *MIR156* genes and repress their expression, thus releasing the downstream SPL transcription factors to regulate a wide range of physiological responses, including branching[24]. In addition, earlier studies reported that mutations in *Arabidopsis FHY3* and *FAR1*, which encode two homologous transcription factors essential for phytochrome A (phyA)-mediated far-red light signaling in *Arabidopsis*[25–27], caused reduced branching number[28]. Despite the progress made in this area, however, how the light signaling pathway integrates with the SL and MIR156/SPL regulatory module to coordinately regulate branching remains poorly understood.

In this study, we report that FHY3 and FAR1 directly interact with SPL9 and SPL15 and inhibit the binding of SPL9 and SPL15 to the *BRC1* promoter, whereas the D53-like proteins SMXL6/7/8 directly interact with SPL9 and SPL15 and suppress their transactivation activity on *BRC1*, thus promoting lateral bud outgrowth and branching. In response to simulated shade conditions, FHY3 and FAR1 protein levels reduce, which on one hand leads to the release of SPL9 and SPL15 to bind to *BRC1* promoter and activate its expression, and on the other hand, leads to reduced expression of *SMXL6* and *SMXL7*, which is directly upregulated by FHY3 and FAR1, resulting in elevated expression of *BRC1* and suppression of branching. Our results establish a complex regulatory network regulating branching under shade conditions through integrating the light and SL signaling pathways with the MIR156/SPL regulatory module.

## Results

**FHY3 and FAR1 regulate branching in response to shade.** Previous studies reported that *fhy3* loss-of-function has reduced lateral branching[28], indicating that FHY3 is a positive regulator of branching. To confirm this, we firstly compared the number of rosette branches (longer than 2 mm) in wild type (WT), *fhy3-11* and *far1-4* single mutants, the *fhy3 far1* double mutant, and the *FHY3* overexpressor (*FHY3-OE*) under normal high R/FR (white light, WL) conditions. Our observation showed that the *FHY3-OE* plants had much more rosette branches (27.3% more than that of WT), while the mutant plants all had less branches compared with the WT (only 16.4% and 13.4% of WT for *fhy3-11* and *fhy3 far1*, respectively) (Fig. 1a, b). Further RT-qPCR analysis showed that the transcript level of *BRC1* was significantly higher in both the *fhy3-11* and *fhy3 far1* plants but obviously lower in the *FHY3-OE* plants compared with the WT plants (Supplementary Fig. 1).

To test whether FHY3 and FAR1 play a role in shade induced repression of branching, we compared the number of rosette branches of WT, *fhy3 far1* double mutant and *FHY3-OE* plants. We treated the plants with far-red light (15 µmol m$^{-2}$ s$^{-1}$) for 30 min at the end of each light period before returning to darkness (EOD-FR) to simulate the shade conditions[29]. Seven-day-old seedlings of WT, *fhy3 far1* double mutant and *FHY3-OE* were treated with EOD-FR for 30 min every day before returning to darkness. After 4 week's treatment, the number of rosette branches (>2 mm in length) was counted. The results showed that compared with their counterparts grown under normal WL conditions, all EOD-FR treated plants had significantly reduced rosette branches (Fig. 1a, b), especially for the WT and *FHY3-OE* plants (~51% and ~62% reduction for WT and *FHY3-OE*, respectively), while the *fhy3-11* and *fhy3 far1* mutants had less decline (~31% and ~27% of reduction for *fhy3-11* and *fhy3 far1*, respectively) (Fig. 1a, b). These observations suggest that the *FHY3-OE* plants are more sensitive, while the *fhy3-11* and *fhy3 far1* mutants are less sensitive to the simulated shade treatment than the WT plants. Consistent with this, RT-qPCR analysis revealed that the transcript level of *BRC1* was significantly upregulated in all shade-treated materials compared with their

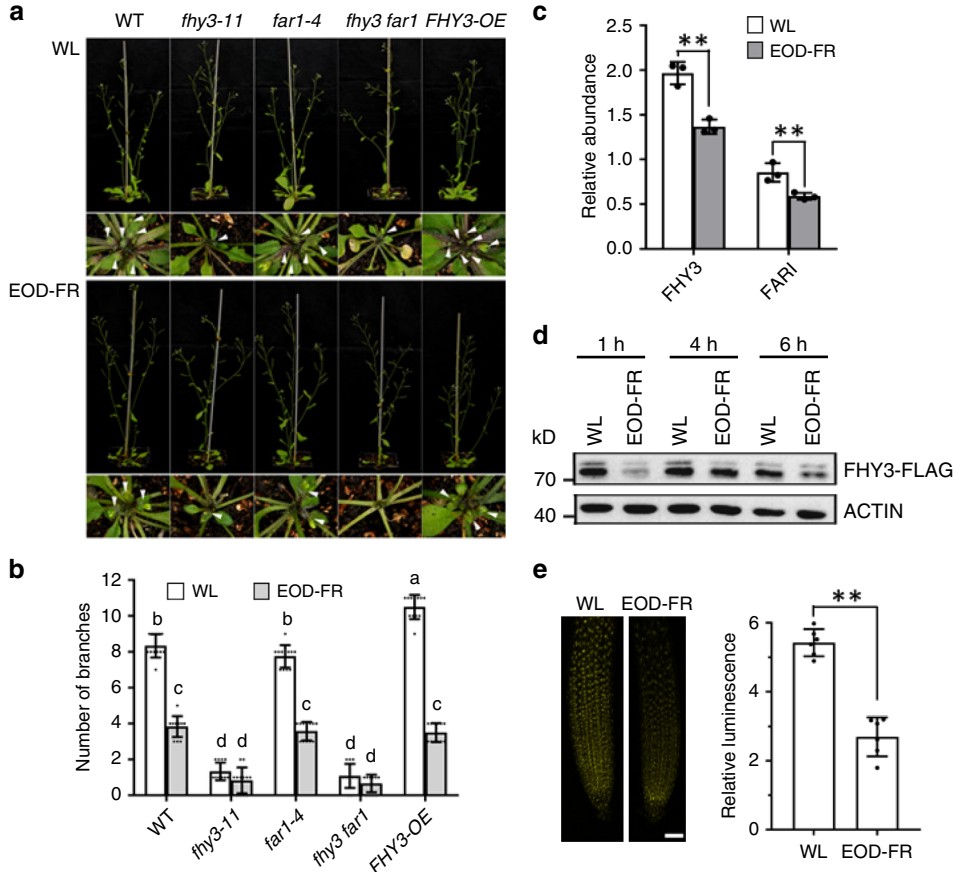

**Fig. 1 FHY3 and FAR1 play a role in regulating branching in response to light quality change. a** Comparison of the rosette branch number of *fhy3-11*, *far1-4*, *fhy3 far1*, *FHY3* overexpressor, and WT plants grown under normal conditions (WL) or stimulated shade conditions (EOD-FR). Eight-day-old seedlings were moved into the soil and grown under WL with or without EOD-FR treatment for 4 weeks before phenotyping. Arrow indicates the short rosette branches. **b** Quantification of the number of rosette branches of *fhy3-11*, *far1-4*, *fhy3 far1*, *FHY3* overexpressor, and WT plants grown under WL or EOD-FR conditions. Values shown are mean ± SD ($n = 12$). Letters indicate significant differences by two-sided LSD test ($p < 0.05$). **c** The transcript levels of *FHY3* and *FAR1* are downregulated by EOD-FR treatment. Seven-day-old seedlings grown under WL conditions were treated with or without EOD-FR for 30 min and then harvested for RNA extraction. Values shown are mean ± SD ($n = 3$). **$p < 0.01$ by two-sided Student *t* test. **d** Western blotting assay showing that both FHY3 protein level rapidly declined in seedlings treated with EOD-FR. Anti-FLAG antibodies were used to detected FHY3 protein and actin was adopted as a loading control. This assay was repeated for three times and similar results were obtained. **e** Fluorescence microscopic analysis of FHY3-YFP protein levels. This assay was repeated for three times and similar results were obtained. The relative fluorescence intensity was quantified by measuring the fluorescence pixel intensity using software Image J. Values shown are mean ± SD ($n = 6$). **$p < 0.01$ by two-sided Student *t* test. Scale bar = 50 μm.

counterparts grown under normal WL conditions, with the highest relative increase of expression being detected in the *FHY3-OE* plants subjected to simulated shade treatment (Supplementary Fig. 1).

We next examined whether the transcript levels of *FHY3* and *FAR1* are regulated by simulated shade treatment. RT-qPCR analysis revealed that, compared with the control plants, the expression of both *FHY3* and *FAR1* were significantly downregulated in plants treated with EOD-FR, with ~38.5% and 35.4% reduction for *FHY3* and *FAR1*, respectively (Fig. 1c). To assess the effect of simulated shade treatment on the protein accumulation of FHY3, 7-day-old *35S::FLAG-FHY3-HA* transgenic seedlings[30] were treated with 15 μmol m$^{-2}$ s$^{-1}$ FR light for 30 min at the end of the light period. The seedlings were harvested at 1, 4, and 6 h post treatment for western blotting analysis. The results showed that accumulation of FHY3 protein rapidly decreased after the EOD-FR treatment (Fig. 1d). Consistent with this, the accumulation of the YFP-FHY3 fusion protein in the *pFHY3::YFP-FHY3* transgenic seedlings[31] also declined in the EOD-FR treated samples (Fig. 1e). We further used proteasome inhibitor MG132

to test whether the decline of FHY3 protein accumulation under simulated shade is regulated by 26S proteasome. Treatment with MG132 significantly slowed down the degradation of FHY3 in the samples treated with simulated shade (Supplementary Fig. 2a, b), indicating that FHY3 protein stability is regulated by the 26S proteasome pathway.

**SPL9/15 inhibit branching by repressing *BRC1* expression.** Previous studies have shown that *Arabidopsis SPL9* and *SPL15* negatively regulate branching and that their loss-of-function mutants display an enhanced branching phenotype[14,15]. To confirm their roles in repressing branching, we generated *spl9 spl15* and *brc1 brc2* double knockout mutants, transgenic overexpressors of *BRC1* (*BRC1-OE*) and *rSPL9* (*rSPL9-OE*, which expresses a miR156-resistant form of *SPL9*) (Supplementary Fig. 3). As expected, the *spl9 spl15* double mutant and the *brc1 brc2* double mutant had more rosette branches than the WT (~39% and ~74.4% more than WT for *spl9 spl15* and *brc1 brc2*, respectively), whereas the *rSPL9-OE* and *BRC1-OE* plants had less

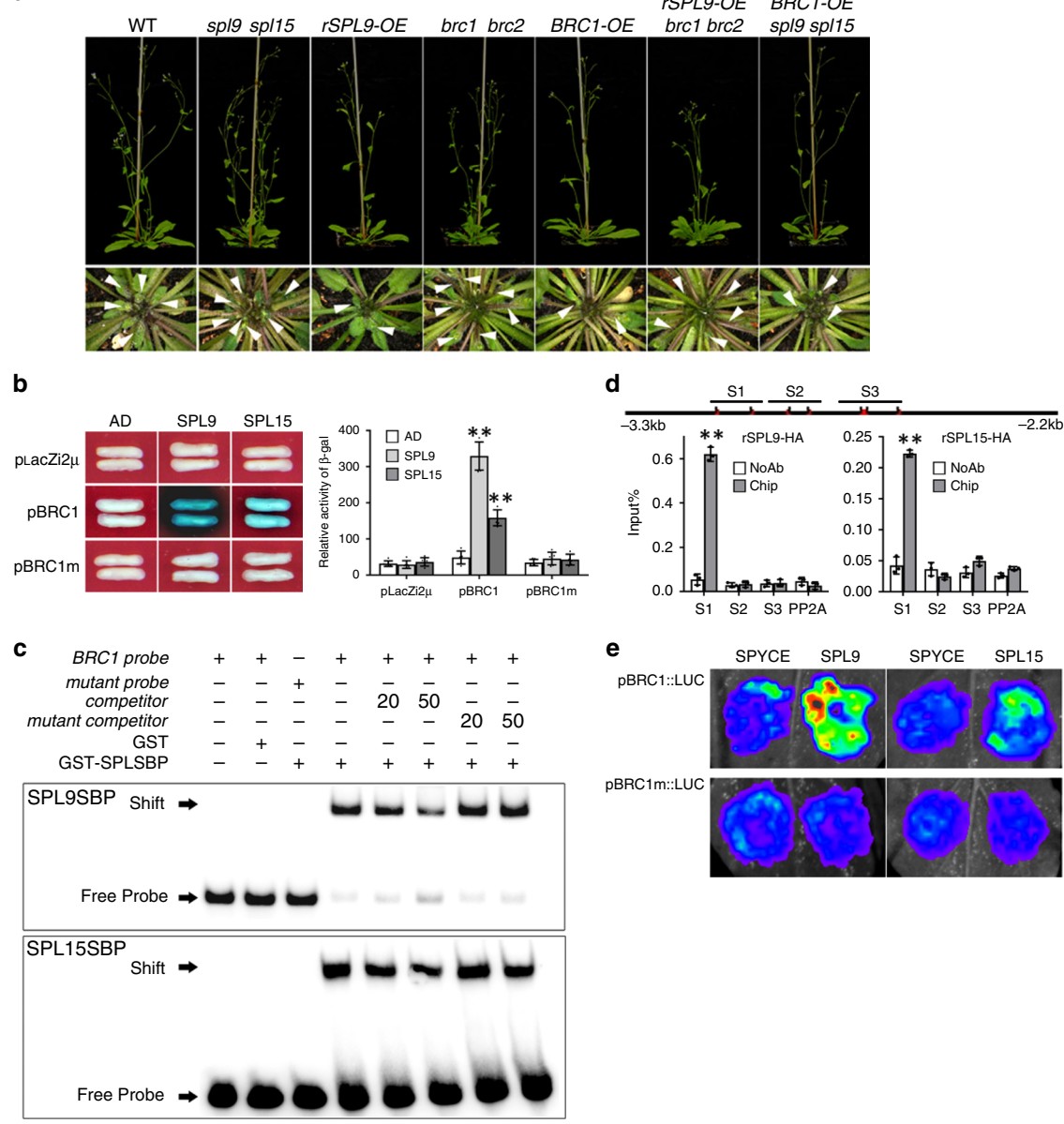

**Fig. 2 SPL9 and SPL15 bind to the *BRC1* promoter and inhibit shoot branching by upregulating *BRC1* expression. a** Comparison of the rosette branch number of WT and various mutants. Eight-day-old seedlings were moved into the soil and grown under normal conditions (WL) for 4 weeks before phenotyping. Arrow indicates the short rosette branches. **b** Yeast one-hybrid assay showing the binding of SPL9 and SPL15 to the full-length promoter (~3.2 kb) of *BRC1*. **c** EMSA shows that both the GST-SPL9SBP domain (up panel) and GST-SPL15SBP domain (down panel) recombinant proteins directly bind to the biotin-labeled probe of the *BRC1* promoter fragment. This assay was repeated for three times and similar results were obtained. **d** ChIP assay of *BRC1* in 10-day-old *35 S::rSPL9-HA* and *35 S::rSPL15-HA* seedlings grown under normal white light conditions. Values given are mean ± SD ($n = 3$). **$p < 0.01$ by two-sided LSD test. **e** Transient expression assay of luminescence intensity show that both SPL9 and SPL15 directly enhance the expression of *BRC1*. Representative images of *N. benthamiana* leaves 72 h after infiltration were shown. Five independent determinations were assessed.

rosette branches than the WT (~72.4% and ~44.2% of WT for *rSPL9-OE* and *BRC1-OE*, respectively) (Fig. 2a and Supplementary Fig. 4). RT-qPCR assay showed that *spl9 spl15* had significantly lower *BRC1* expression level than WT, while *rSPL9-OE* had significantly higher *BRC1* expression level than WT (Supplementary Fig. 5). To test whether *SPL9* and *SPL15* repress branching through *BRC1* and *BRC2*, we generated the *rSPL9-OE/brc1 brc2* and *BRC1-OE/spl9 spl15* high-order mutants via genetic crosses. Phenotypic analysis showed that *rSPL9-OE/brc1 brc2* had similar number of branches as the *brc1 brc2* double mutant, while *BRC1-OE/spl9 spl15* had similar branch number as the *BRC1-OE* plants (Fig. 2a and Supplementary Fig. 4).

RT-qPCR assay showed that *brc1 brc2/rSPL9-OE* had similar *BRC1* expression level as *brc1 brc2*, whereas *BRC1-OE/spl9 spl15* had similar *BRC1* expression level as *BRC1-OE* (Supplementary Fig. 5). These observations support the notion that *SPL9* and *SPL15* repress branching through regulating the expression of *BRC1*.

As previous studies have shown that rice IPA1 can directly bind to the promoter of *FC1* and activate its expression, and that bioinformatics analysis revealed the presence of GTAC motifs for SPL protein binding in the *BRC1* and *BRC2* promoters (Supplementary Fig. 6), we speculated that *Arabidopsis* SPL9 and SPL15, which are most closely related to IPA1 based on a

phylogenetic analysis[13], may also directly bind to the *BRC1* and *BRC2* promoters and regulate their expression. Yeast one-hybrid assay showed that both SPL9 and SPL15 could bind to the S1 fragment (which contains two GTAC motifs) of the *BRC1* promoter, but not the *BRC2* promoter. Mutation of the GTAC core motifs in the S1 fragment of the *BRC1* promoter into TTTT abolished the binding, suggesting that the binding is specific (Fig. 2b and Supplementary Fig. 6). To confirm the binding, we performed electrophoresis mobility shift assay (EMSA) using purified recombinant proteins of the SBP domain of SPL9 and SPL15 fused with the glutathione S-transferase (GST) tag (Supplementary Fig. 7), and the results showed that both SPL9 and SPL15 could bind to the biotin-labeled probe of *BRC1* (Fig. 2c). Further, we performed chromatin immunoprecipitation (ChIP)-qPCR assay using transgenic plants that constitutively express miR156-resistant forms of SPL9 and SPL15 fused with HA tag at their C terminal (*35S::rSPL9-HA* and *35S::rSPL15-HA*, respectively). As shown in Fig. 2d, the S1 fragment of the *BRC1* promoter was significantly enriched in the ChIP samples. Together, these observations indicate that SPL9 and SPL15 can directly bind to the *BRC1* promoter.

Next, we carried out a transient expression assay in *N. benthamiana* leaves to examine the effect of SPL9 and SPL15 on *BRC1* transcription. The results showed that the p*BRC1::LUC* reporter activity was significantly elevated when co-transfected with either the *35S::rSPL9* or the *35S::rSPL15* construct, compared with the control vector (pSPYCE) (Fig. 2e). To verify the physiological relevance of SPL9/15 activation of *BRC1* expression in planta, we generated transgenic plants expressing *BRC1* driven by its endogenous WT promoter (*pBRC1wt::gBRC1*) or *BRC1* promoter mutated in the SBP-binding site (*pBRC1m::gBRC1*) in the *spl9 spl15/brc1 brc2* and *rSPL9-OE/brc1 brc2* backgrounds. We found that the branch number in the *brc1 brc2* plants harboring the *pBRC1wt::gBRC1* transgene was restored to WT level, whereas the *brc1 brc2* plants harboring the *pBRC1m::gBRC1* transgene had significantly more branches than the WT plants (similar to *brc1 brc2*). We also found that the *spl9 spl15/brc1 brc2* plants carrying the *pBRC1wt::gBRC1* or *pBRC1m::gBRC1* transgenes had similar branch number as the *spl9 spl15/brc1 brc2* plants (Supplementary Fig. 8). Moreover, we found that *rSPL9-OE/brc1 brc2* plants harboring *pBRC1wt::gBRC1* transgene had significantly fewer branches than *brc1 brc2* and WT plants. However, no significant difference was detected between *rSPL9-OE/brc1 brc2* plants harboring the *pBRC1m::gBRC1* transgene and the *brc1 brc2* mutant plants (Supplementary Fig. 8). Collectively, these results strongly support the notion that *SPL9* and *SPL15* act to regulate branch number through regulating *BRC1* expression via binding to the SBP core motif in the *BRC1* promoter.

**FHY3 and FAR1 directly interact with SPL9 and SPL15**. As FHY3 and FAR1 play an antagonistic role with SPL9 and SPL15 in regulating branching, we wondered whether these proteins physically interact with each other to coordinately regulate *BRC1* expression. Yeast two-hybrid assay showed that both FHY3 and FAR1 interacted with both SPL9 and SPL15 (Fig. 3a). Domain deletion analysis revealed that both the FHY3-I (N-terminal domain containing the DNA-binding zinc finger motif) and FHY3-II domains, but not the FHY3-III domain, could interact with SPL9 and SPL15 (Fig. 3a). The interaction between FHY3 and FAR1 with SPL9 and SPL15 was further confirmed using bimolecular fluorescence complementation (BiFC) assay and luciferase complementation imaging (LCI) assay in *N. benthamiana* leaf epidermis, in vitro pull-down assay and in vivo co-immunoprecipitation (Co-IP) assay (Fig. 3b–d and Supplementary Fig. 9). Domain mapping analysis also revealed

that the SBP-box domain of SPL9 and SPL15 is responsible for interacting with FHY3 and FAR1 (Supplementary Fig. 10).

**FHY3 and FAR1 inhibit SPL9/15 binding to the *BRC1* promoter**. We next tested how the interaction between FHY3 and FAR1 with SPL9 and SPL15 may affect the transcriptional regulation of *BRC1*. Yeast one-hybrid assay showed that inclusion of an *FHY3*-expressing construct in the yeast obviously inhibited the activation of SPL9 and SPL15 on the *BRC1* promoter-driven reporter gene (Fig. 4a), suggesting that the interaction between FHY3 and SPL9/15 may inhibit the binding of SPL9 and SPL15 to the *BRC1* promoter. To confirm this, we carried out EMSA by adding the FHY3 protein (FHY3-I domain) to the mixture of SPL-SBP and biotin-labeled *BRC1* probe. The results showed that FHY3 protein itself could not bind to the *BRC1* probe, but addition of FHY3 protein (but not the MBP control) significantly reduced the amounts of shifted probes caused by binding of SPL9 and SPL15 (Fig. 4b). To further verify the inhibitory role of FHY3 on the DNA-binding activity of SPL9 and SPL15 in vivo, we performed ChIP-qPCR assay using WT, *FHY3-OE* and *fhy3 far1* double mutants carrying the same *35S:rSPL9-HA* transgene grown under normal white light conditions or subjected to EOD-FR treatment. Compared with the WT background, the enrichment of *BRC1* DNA fragment declined drastically in the *FHY3-OE* background, but increased in the *fhy3 far1* double mutant background (Fig. 4c). In addition, the enrichment of *BRC1* DNA fragment was significantly higher in the EOD-FR treated WT and *FHY3-OE* plants compared with their counterparts grown under normal white light conditions (Fig. 4c and Supplementary Fig. 11). Together, these results strongly support the notion that FHY3 can inhibit the DNA-binding activity of both SPL9 and SPL15 to the *BRC1* promoter and that SPL9 (and probably SPL15 as well) has stronger binding activity to the *BRC1* promoter under simulated shade conditions.

Next, we performed a transient expression assay to test the effect of FHY3 on the transcriptional activation activity of SPL9 and SPL15 on *BRC1* by co-infiltrating the *35S::rSPL* construct and *pBRC1::LUC* reporter with or without *35S::FHY3* into *N. benthamiana* leaves. As shown in Fig. 4d, expression of the reporter gene declined significantly in the samples co-injected with *35S::FHY3*, suggesting that FHY3 indeed represses the transcriptional activation activity of SPL9 and SPL15 on *BRC1*.

To provide genetic evidence supporting the above notion, we generated *fhy3 far1/spl9 spl15* and *FHY3-OE/spl9 spl15* mutant combinations via genetic crosses and compared their rosette branch numbers. The results showed that both *fhy3 far1/ spl9 spl15* and *FHY3-OE/spl9 spl15* had similar number of rosette branches as the *spl9 spl15* double mutant (Fig. 4e, f). RT-qPCR assay showed that *fhy3 far1/spl9 spl15* and *FHY3-OE/spl9 spl15* had similar *BRC1* expression levels as *spl9 spl15* (Supplementary Fig. 12). These results together support the notion that *FHY3* and *FAR1* indeed act upstream of *SPL9* and *SPL15* to promote rosette branching.

**SMXL6/7/8 interact with SPL9/15**. Previous studies also showed that in rice, D53 (homolog of SMXL6/7/8) can physically interact with IPA1 (homolog of SPL9 and SPL15) and suppress the transcriptional activity of IPA1[19]. We thus tested whether SMXL6/7/8 can interact with SPL9 and SPL15. As expected, our yeast two-hybrid assay showed that all these three SMXL proteins could interact with both SPL9 and SPL15, with stronger interaction being observed for SPL9 (Fig. 5a). The interaction between SMXL6/7/8 and SPL9/15 was further verified by LCI assay and BiFC assay (Fig. 5b and Supplementary Fig. 13). We further selected SMXL6 as a test case and confirmed its interaction with

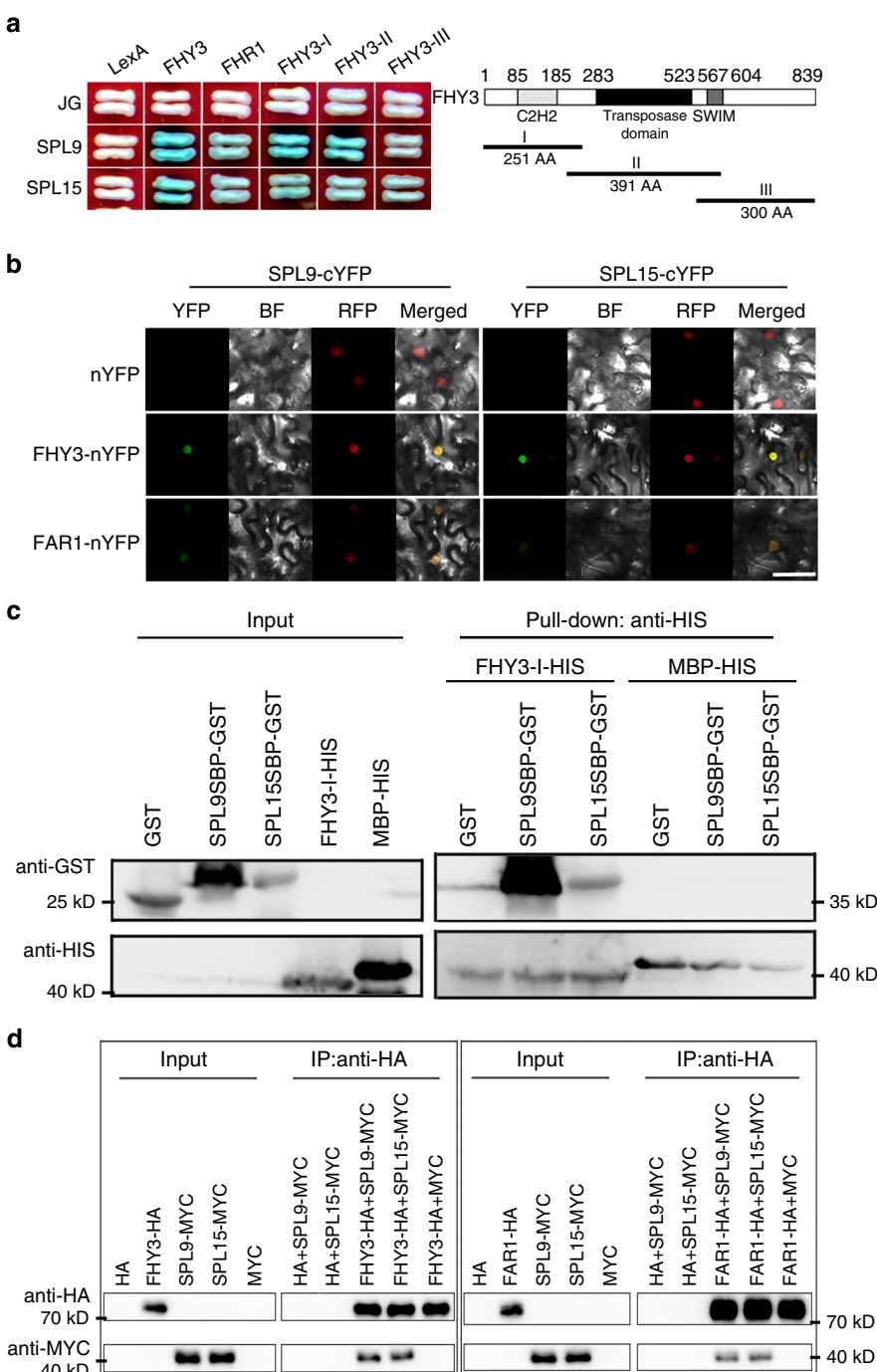

**Fig. 3 FHY3 and FAR1 directly interact with SPL9 and SPL15. a** Yeast two-hybrid assay shows that both SPL9 and SPL15 interact with FHY3 and FAR1. LexA and JG serve as the negative controls. **b** BiFC assay shows that both SPL9 and SPL15 interact with FHY3 and FAR1 in *N. benthamiana* leaf epidermal cells. Both FHY3 and FAR1 were fused to the N-terminal fragment of YFP (nYFP) and both SPL9 and SPL15 were fused to the C-terminal fragment of YFP (cYFP). The AT-HOOK-RFP marker was used to indicate the nuclei. The interaction between nYFP and SPL9-cYPF or SPL15-cYPF serves as the negative controls. Scale bar = 50 μm. **c** Pull-down assay shows that FHY3 directly interacts with both SPL9 and SPL15 in vitro. GST-SPL9SBP or GST-SPL15SBP proteins were incubated with protein extracts containing His-FHY3-I and further immobilized with glutathione sepharose beads. **d** Co-IP analysis shows the interaction between FHY3/FAR1 and SLP9/15 using a transient expression system in tobacco leaves. Immunoprecipitation was performed using anti-HA matrix and the co-immunoprecipitated proteins were detected with anti-MYC antibodies. This assay was repeated for three times and similar results were obtained.

SPL9 and SPL15 using in vitro pull-down assay and in vivo Co-IP assay (Fig. 5c, d).

We next investigated whether SMXL proteins could affect the DNA-binding activity of SPL9 and SPL15 to the *BRC1* promoter. Both yeast one-hybrid assay and EMSA assay showed that no

obvious effect was observed when SMXL6 was added into these assays (Fig. 5e and Supplementary Fig. 14), suggesting that SMXL proteins are unlikely to affect the DNA-binding activity of SPL9 and SPL15. Next, we performed transient expression assay in *N. benthamiana* leaves to test the effect of SMXL proteins on

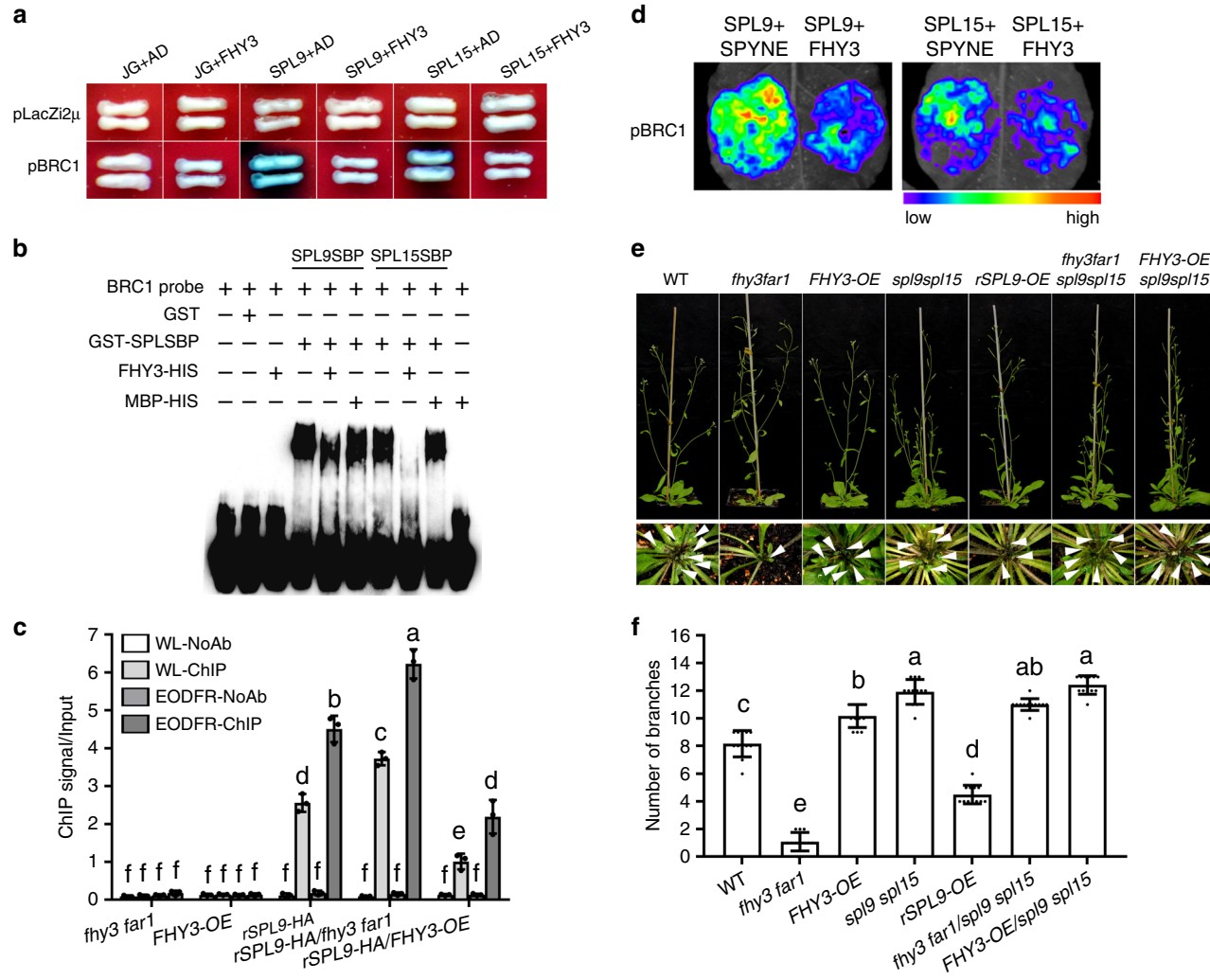

**Fig. 4 FHY3 inhibits the binding and transactivation function of SPL9 and SPL15. a** Yeast one-hybrid assay shows that FHY3 inhibits the binding of SPL9 and SPL15 to the *BRC1* promoter. **b** EMSA assay shows that FHY3 inhibits the direct binding of SPL9SBP and SPL15SBP to the *BRC1* promoter. MBP protein is used as the negative control. This assay was repeated for three times and similar results were obtained. **c** ChIP assay of *BRC1* in axillary buds from 4-week-old *35S::rSPL9-HA* transgenic plants in the WT, *fhy3 far1*, or *FHY3-OE* backgrounds. Values given are mean ± SD ($n = 3$). Letters indicate significant differences by two-sided LSD test ($p < 0.05$). **d** Transient expression assay and quantitative analysis of luminescence intensity show that FHY3 inhibits the transactivation activity of both SPL9 and SPL15 on the *LUC* reporter gene driven by the *BRC1* promoter. Representative images of *N. benthamiana* leaves 72 h after infiltration were shown. Five independent determinations were assessed. **e** Comparison of the rosette branch number of 4-week-old *fhy3 far1*, *spl9 spl15*, *FHY3-OE*, *rSPL9-OE*, and their higher order mutants grown under normal white light conditions. Arrow indicates the short rosette branches. **f** Quantification of the number of rosette branches of *fhy3 far1*, *spl9 spl15*, *FHY3-OE*, *rSPL9-OE*, and their higher order mutants grown under normal white light conditions. Values shown are mean ± SD ($n = 12$). Letters indicate significant differences by two-sided LSD test ($p < 0.05$).

transcriptional activation of *BRC1* expression. The results showed that addition of SMXL6 protein significantly reduced the activation activity of SPL9 and SPL15 on the reporter gene driven by the *BRC1* promoter, compared with the addition of the pSPYNE empty vector. Similar results were also observed for the SMXL7 and SMXL8 proteins (Fig. 5f). These results demonstrate that SMXL6/7/8 inhibit the transcriptional activation activities of both SPL9 and SPL15 on *BRC1*.

To provide genetic evidence for the above notion, we generated *smxl6/7/8* triple mutant and *SMXL6D* (a mutant form of SMXL6 resistant to degradation) overexpressors (*SMXL6D-OE*) as described[11] (Supplementary Fig. 15). As expected, the *smxl6/7/8* triple mutant had fewer, while the *SMXL6D* over-expressors had more branches than the WT (Fig. 6a, b). We next generated *smxl6/7/8/spl9/15* quintuple mutant and *SMXL6-OE/ rSPL9-OE* combinations. Phenotypic assay showed that the

*smxl6/7/8/spl9/15* quintuple mutant had similar number of rosette branches as *spl9 spl15*, while *SMXL6D-OE/ rSPL9-OE* had slightly more (~15.7%) rosette branches than *rSPL9-OE* (Fig. 6a, b). RT-qPCR assay showed that *smxl6/7/8/spl9/15* had similar *BRC1* transcript level as *spl9/15*, while *SMXL6D-OE/ rSPL9-OE* had similar *BRC1* transcript level as *rSPL9-OE* (Supplementary Fig. 16). Together, these results suggest that *SMXLs* act to promote rosette branching through suppressing the transcriptional activation activities of both SPL9 and SPL15 on *BRC1*.

**FHY3 and FAR1 directly upregulate the expression of *SMXL6/ 7*.** As we observed direct interaction between SPL9 and SPL15 with FHY3/FAR1 and SMXL proteins, we wondered whether FHY3 and FAR1 may interact with SMXL proteins as well. Our

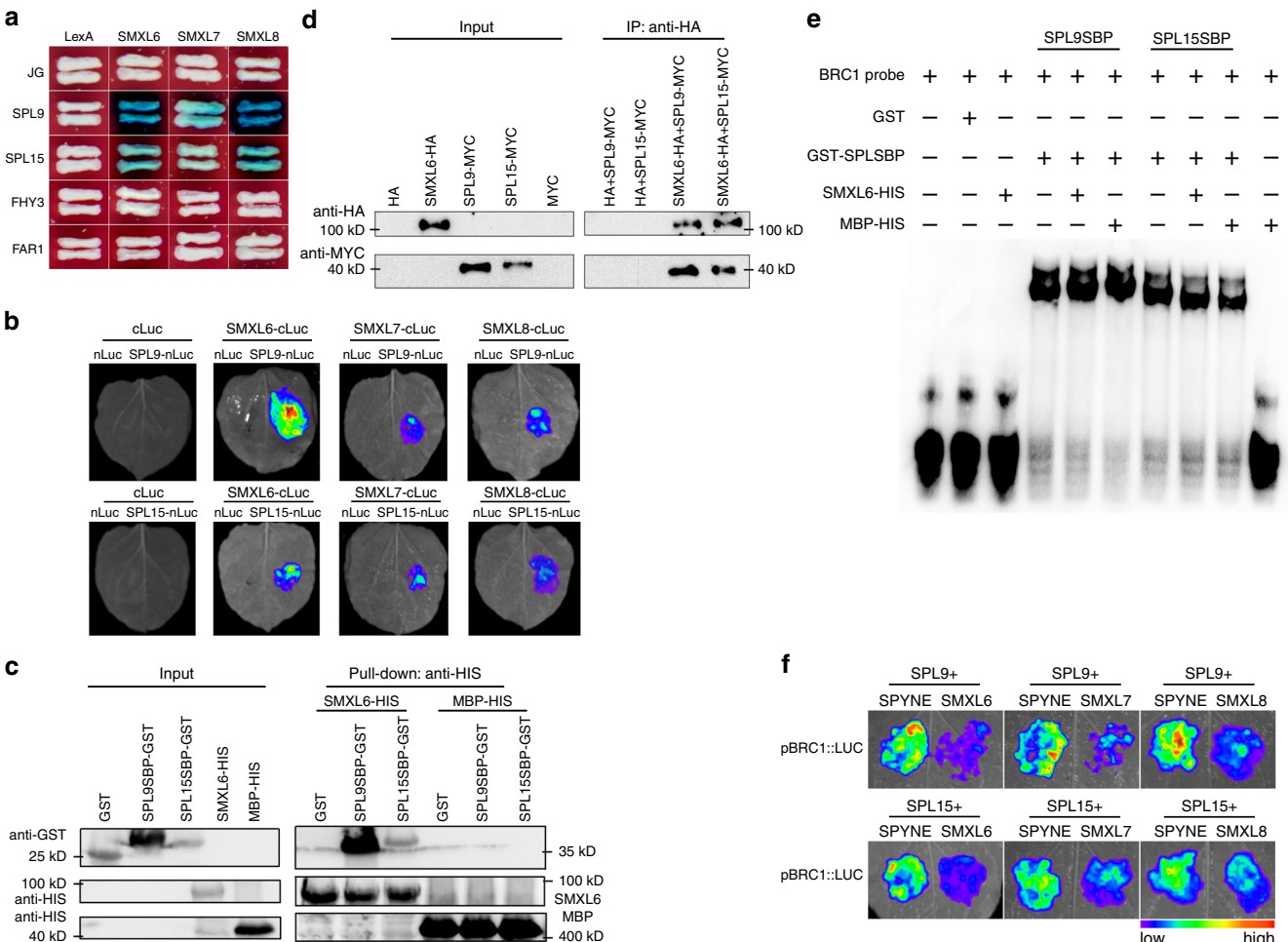

**Fig. 5 SMXL6, SMXL7, and SMXL8 directly interact with SPL9 and SPL15. a** Yeast two-hybrid assay shows that SMXL6, SMXL7, and SMXL8 interact with SPL9 and SPL15. LexA and JG served as the negative controls. **b** LCI assay and quantitative analysis of luminescence intensity showing the interaction between SPL9/SPL15 and SMXL6/7/8 in *N. benthamiana* epidermal cells. SPL9 and SPL15 were fused to the N-terminal fragment of luciferase (nLuc) while SMXL6/7/8 were fused to the C-terminal fragment of luciferase (cLuc). The interactions between cLuc and SPL9-nLuc or SPL15-nLuc were used as negative controls. Representative images of *N. benthamiana* leaves 72 h after infiltration were shown. Four independent determinations were assessed. **c** Pull-down assay shows that SMXL6 directly interacts with both SPL9 and SPL15 in vitro. Purified GST-SPL9SBP or GST-SPL15SBP recombinant proteins were incubated with protein extracts containing His-SMXL6 and further immobilized with glutathione sepharose beads. **d** Western blot of immunoprecipitated proteins (left panel) and co-immunoprecipitated proteins (right panel) transiently expressed in *N. benthamiana*. Immunoprecipitation was performed using anti-HA matrix and the co-immunoprecipitated proteins were detected with anti-MYC antibodies. **e** EMSA assay shows that SMXL6 protein does not inhibit the binding of SPL9SBP and SPL15SBP to the *BRC1* promoter fragment. MBP protein is used as the negative control. **f** Transient expression assay shows that SMXL6, SMXL7, and SMXL8 repress the transactivation activity of both SPL9 and SPL15 on the *LUC* reporter gene driven by the *BRC1* promoter.

yeast two-hybrid assay showed that no direct interaction could be detected between FHY3/FAR1 and SMXL6/7/8 (Fig. 5a). Notably, bioinformatics analysis identified at least two putative FHY3/FAR1-binding sites (FBS, CACGCGC) in the first exon of both *SMXL6* and *SMLX7* (Supplementary Fig. 17). Thus, we tested whether *SMXL6* and *SMXL7* are direct target genes of FHY3. Yeast one-hybrid assay showed that FHY3 could directly bind to the first exon of *SMXL6* and *SMXL7* (Fig. 6c). Moreover, transient expression assay in *N. benthamiana* leaves showed that co-expression of FHY3 significantly enhanced the transcription of *LUC* reporter genes driven by the 35S minimal promoter fused with the 5′ UTR together with the first exon of *SMXL6* and *SMXL7* (Fig. 6d and Supplementary Fig. 18). Further, RT-qPCR assay showed that the transcript levels of *SMXL6/7* were all downregulated in the *fhy3 far1* double mutant, but upregulated in the *FHY3-OE* plants (Fig. 6e). In addition, the expression levels of *SMXL6* and *SMXL7* significantly declined in the WT and

*FHY3-OE* plants grown under simulated shade conditions, compared with their counterparts under normal white light conditions (Fig. 6e). These results suggest that FHY3 (and possibly FAR1) can directly upregulate the expression of *SMXL6* and *SMXL7*, and thus likely play a role in SL signaling.

To test whether *FHY3/FAR1* and *SMXL6/7* could directly regulate *BRC1* expression, we first examined whether FHY3/FAR1 and SMXL6/7 can directly bind to the *BRC1* promoter. Yeast one-hybrid assay revealed that there was no direct binding of FHY3/FAR1 or SMXL6/7 to the *BRC1* promoter (Supplementary Fig. 19). Secondly, we tested whether FHY3/FAR1 and SMXL6/7 can regulate BRC1 activity through protein–protein interaction. Our yeast two-hybrid assay showed that no direct interaction was detected between BRC1 and FHY3/FAR1 or SMXL6/7 proteins (Supplementary Fig. 20). These results suggest that it is unlikely *FHY3/FAR1* and *SMXL6/7* directly regulate *BRC1* expression or BRC1 activity.

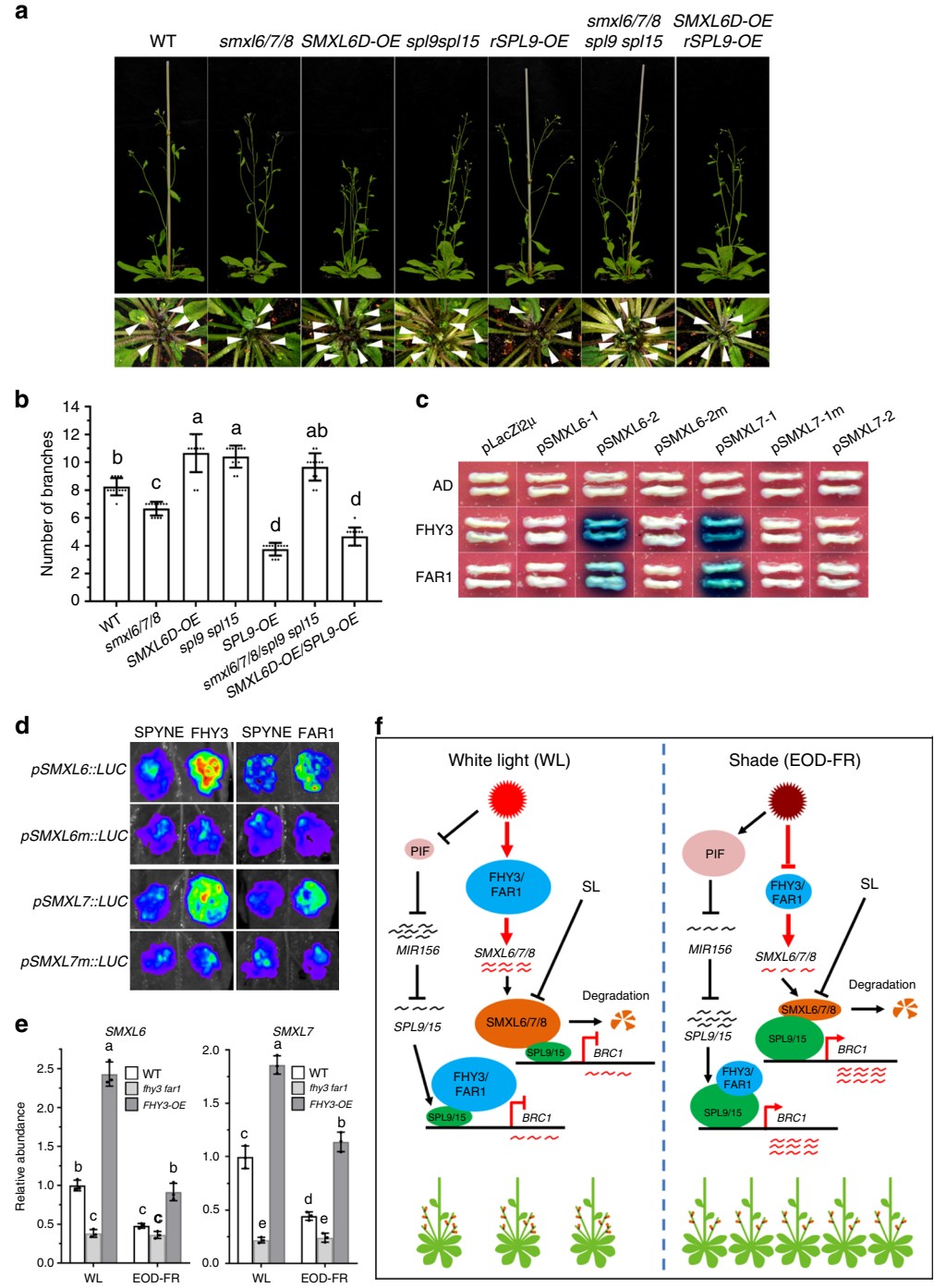

**Fig. 6 FHY3 and FAR1 directly upregulate the expression of *SMXL6* and *SMXL7*. a** Comparison of the rosette branch number of *smxl6/7/8*, *spl9 spl15*, *SMXL6D-OE*, *rSPL9-OE*, and their higher order mutants grown under normal conditions. Eight-day-old seedlings were moved into the soil and grown under normal conditions for 4 weeks before phenotyping. Arrow indicates the short rosette branches. **b** Quantification of the number of rosette branches of *fhy3 far1*, *spl9 spl15*, *FHY3-OE*, *rSPL9-OE*, and their higher order mutants. Values shown are mean ± SD (*n* = 12). Letters indicate significant differences by two-sided LSD test (*p* < 0.05). **c** Yeast one-hybrid assay shows that FHY3 and FAR1 bind to the promoters of *SMXL6* and *SMXL7*. pLacZi2μ and JG served as the negative controls. **d** Transient expression assay shows that FHY3 and FAR1 promote the expression of *SMXL6* and *SMXL7*. Representative images of *N. benthamiana* leaves 72 h after infiltration were shown. **e** Comparison of the expression levels of *SMXL6* and *SMXL7* in 7-day-old wild type, *fhy3 far1* and *FHY3-OE* seedlings grown under normal white light or simulated shade (EOD-FR) conditions. Values given are mean ± SD (*n* = 3). Different letters indicate significant differences by two-sided LSD test (*p* < 0.05). **f** A simplified schematic model depicting FHY3 and FAR1 integrate light and strigolactone signaling to regulate branching. FHY3/FAR1 and SMXL6/SMXL7/SMXL8 directly interact with SPL9 and SPL15 and suppress their transcriptional activation of *BRC1*, meanwhile FHY3 and FAR1 upregulate the expression levels of *SMXL6* and *SMXL7* and thus promote branching. Simulated shade treatment reduces the accumulation of FHY3 protein, leading to increased expression of *BRC1* and reduced branching. Arrow: activate; Bar: repress.

## Discussion

Previous studies have shown that light and SLs co-regulate many developmental and physiological processes in plants. For example, it has been shown that SL-mediated inhibition of hypocotyl elongation is dependent on phytochrome and cryptochrome signaling pathways[32,33]. In addition, earlier studies showed that *MAX2*, which encodes a subunit of an SCF E3 ligase, acts as a key positive regulator of the SL signaling pathway and inhibits shoot branching, also functions positively in regulating seedling photomorphogenesis and seed germination[34,35]. Moreover, it has been reported that high R/FR conditions promote the biosynthesis of SLs, while low R/FR conditions inhibit the biosynthesis of SLs in tomato and *Lotus japonicus*[36,37]. These results collectively suggest that light and SLs coordinately regulate multiple developmental process in an interdependent manner.

In this study, we found that FHY3 and FAR1, two crucial components of phytochrome A (phyA)-mediated far-red light signaling in *Arabidopsis*, could directly interact with SPL9 and SPL15 in vivo and in vitro, and inhibit their binding to the promoter of the branching integrator *BRC1*. We also found that SMXL6/7/8, three components of the SL signaling pathway, directly interact with SPL9 and SPL15 and suppress their transactivation activities on *BRC1*. Moreover, FHY3 and FAR1 can directly upregulate the expression of *SMXL6* and *SMXL7*. Under stimulated shade conditions, the protein levels of FHY3 and FAR1 decline and the transcript levels of both *SMXL6* and *SMXL7* are downregulated. Based on these results, we propose a putative model in which accumulation of FHY3 and FAR1 proteins decline under stimulated shade conditions, which in turn releases SPL9 and SPL15 proteins from the inhibitory interactions with FHY3/FAR1 and SMXL6/7/8, leading to increased expression of *BRC1* and thus reduced branching (Fig. 6f). Our results suggest that *Arabidopsis FHY3* and *FAR1* act at the nexus of light and SL signaling integration to coordinately regulate branching, thus greatly deepening our understanding of the signaling cross talk mechanisms of light and SL in regulating plant branching and expanding the functional repertoire of FHY3 and FAR1 in regulating plant growth and development.

It is worth mentioning that our previous studies have shown that in response to phytochrome B inactivation under shade conditions, the protein levels of PIFs rapidly increase, and they can directly bind to the promoters of multiple *MIR156* genes and repress their expression, thus releasing the downstream *SPLs* target genes to regulate various aspects of shade avoidance responses, including reduced branching[24]. In this study, we showed that FHY3 and FAR1 interact with both SPL9 and SPL15 and inhibit their binding to the *BRC1* promoter. These results suggest that multiple signaling pathways could simultaneously operate downstream of phytochromes to regulate shade avoidance response. It will be interesting to investigate whether and how the FHY3/FAR1-medited pathway intersects with the PIFs mediated pathway to generate a coherent response in future studies. In this regard, it is interesting to note that bioinformatics analysis revealed that in *Arabidopsis*, the promoters of *MIR156B*, *SPL3*, *SPL7*, *SPL9*, and *SPL10* contain at least one FHY3/FAR1-binding site (FBS, CACGCGC), hinting a probability that *FHY3* and *FAR1* may also participate in shade avoidance response through transcriptional regulation of those factors. Besides, the interaction between DELLA proteins (key repressors of the GA signaling pathway) with SPL9 to coordinately regulate floral transition was recently reported[38]. The possibility of FHY3 and FAR1 in mediating cross talk between light and GA signaling pathway will also be an interesting avenue for future research. Moreover, earlier studies reported that FHY3 promotes branching through an AXR1-dependent pathway[28]. However, the detailed mechanism still remains unknown and will be worthy of further investigation.

The continuous increase in world population and reduced arable land, accompanied by continuously deteriorated climate conditions poses a significant challenge to secure global food supply in the coming decades[39]. Increasing planting density has been an effective means of increasing crop yield per unit land area[40,41]. However, shade avoidance response, which is triggered in response to vegetative proximity or canopy shade, is detrimental to crop productivity. Emerging evidence supports the notion that phytochrome signaling pathway has been target of selection during crop domestication and genetic improvement[42]. In addition, it has been shown that in response to shade, inactivation of phytochrome B leads to increased *TB1* expression and suppression of axillary bud outgrowth in sorghum[43,44] and that maize *grassy tillers1* acts downstream of phyB to suppress axillary bud outgrowth in response to shade signals[45]. Further, the *SPL* gene family has been proposed to be promising targets of genetic manipulation to improve plant architecture and other agronomic traits tailored for high-density planting[12,13]. Given the conserved TB1/FC1/BRC1 pathway in regulating branching/tillering in monocots and dicots, the results described in this report should have important implications for future design of crops with improved plant architecture and thus increased yields, particularly under high-density planting conditions.

## Methods

**Plant material and growth conditions**. The *Arabidopsis thaliana* ecotype Columbia-0 (Col-0) was used in this study. The mutants *spl9-4* (SAIL_150_B05), *spl15-1* (SALK_074426), *fhy3-11* (SALK_002711), *far1-4* (SALK_031652), *brc1-2* (SALK_091920), and *brc2-1* (SALK_023116) were obtained from the Nottingham *Arabidopsis* Stock Centre. The T-DNA insertion sites of these mutant alleles were shown in Supplementary Fig. 21. After surface sterilization, seeds were sown on half strength Murashige and Skoog media plates supplemented with 1% sucrose. After a 3-day stratification period in the dark at 4 °C, the plates were then placed in a growth chamber (Percival, USA) at 23 °C with a 16-h-light/8-h-dark photoperiod to allow seed germination and seedling growth.

*N. benthamiana* seeds were directly sown into soil, germinated in darkness for 4 days, and grown in a culture room at 25 °C with a 16-h-light/8-h-dark photoperiod (light intensity of 100 μmol m$^{-2}$ s$^{-1}$).

**EOD-FR treatment**. For simulated shade treatment, seedlings were treated with 15 μmol m$^{-2}$ s$^{-1}$ FR light at the end of the light period (EOD-FR) for 30 min every day until harvesting or phenotypic examination.

**Rosette branches measurement**. For phenotypic investigation, axillary bud and branches were determined as described in Aguilar-Martinez et al.[5] and Stirnberg et al.[28]. At least 12 individuals for each genotype were grown under normal conditions or simulated shade conditions for 4 weeks after germination and then the number of rosette branches (longer than 2 mm) was counted.

**Plasmid construction**. For yeast one-hybrid assay, the full-length coding sequence of *SPL9* or *SPL15* was PCR amplified from WT cDNA and ligated to the pJG4-5 vector (Clontech, USA) to produce the *AD-SPL9* and *AD-SPL15* constructs. Promoter fragments (~3 kb in length) of *BRC1* or *BRC2* were amplified from WT genomic DNA and ligated to the pLacZi2μ vector[27] to generate the *2 μ-pBRC1* or *2 μ-pBRC2* constructs. For yeast two-hybrid assay, the coding regions of *FHY3*, *FAR1*, or *SMXLs* were amplified and cloned into the pEG202 vector (Clontech, USA) at the *EcoR*I restriction site to generate the corresponding LexA fusion construct.

For BiFC, the coding sequences of *SPL9* or *SPL15* (without stop codon) were cloned into the pSPYCE vector[46] to generate the *35S::SPL-SPYCE* constructs. The coding regions of *FHY3*, *FAR1*, and *SMXLs* (without stop codon) were cloned into the pSPYNE vector to produce the corresponding SPYNE fusion constructs. For LCI assay, the coding sequences of *SPL9* or *SPL15* were cloned into the pCAMBIA1300-cLUC vector and the coding sequences of *FHY3*, *FAR1*, or *SMXLs* were cloned into the pCAMBIA1300-nLUC vector[47] to generate the corresponding constructs. For luciferase assay, the promoter of *BRC1* was cloned into the pGreenII0800-LUC vector (Biovector, USA) at the *Sal*I restriction site to produce the *pBRC1::LUC* construct. All these constructs were introduced into the *A. tumefaciens* strain EHA105.

To generate the *FHY3* and *BRC1* overexpression constructs, the full-length coding sequences of *FHY3* and *BRC1* were cloned into the pCAMBIA1301 vector (Biovector) to produce the *FHY3-OE* and *BRC1-OE* constructs, respectively. For *rSPL9* overexpression, a miR156-resistant form of SPL9[48] without stop codon was cloned into the pSPYCE vector at the *Sal*I restriction site to generate the *rSPL9-OE*

construct. For *SMXL6D* (a nondegradable mutant form of SMXL6[11]) overexpression, *SMXL6D* was constructed into the pSPYNE vector.

To generate the *smxl6/7/8* triple knockout mutant, the CRISPR/Cas9 system was adopted following the method described by Zhao et al[49]. In brief, potential target sites on the exons of *SMXL6, SXML7,* and *SMXL8* were analyzed and selected using the SnapGene® Viewer 3.2 software by the criteria of 5′-GG-(N)18-NGG-3′. The different forward primers sgR-F1 and sgR-F2 were paired with the reverse primer sgR-R to produce sgRNA1 and sgRNA2 fragments. Then these sgRNA fragments driven by the *Arabidopsis* ubiquitin U6-1 promoter were cloned into the CPB vector[49].

To generate the *pBRC1wt::gBRC1* construct which harboring the *BRC1* genomic coding region driven by the WT *BRC1* promoter, the fragment containing the *BRC1* promoter and the coding region was amplified from WT genomic DNA and ligated into the pCAMBIA1300 vector (Biovector) at the *EcoRI/Hind*III restriction sites. The *pBRC1m::gBRC1* construct was produced by replacing the WT *BRC1* promoter with the mutated promoter which is amplified from the *pBRC1wt::gBRC1* construct in which the two SBP-binding sites GTAC were mutated into TTTT via site-directed mutagenesis.

All sequence confirmed transforming vectors were introduced into *A. tumefaciens* strain GV3101 and further transformed into WT *Arabidopsis* or specified backgrounds using the floral dip method[50]. For overexpression vectors, more than 20 independent transgenic lines for each transformation vector were obtained and 3–5 representative lines (T₃ generation, homozygous plants) of each vector were selected for gene expression and phenotyping assays. For the *smxl6/7/8* CRISPR knockout mutant, 27 independent transgenic lines were obtained and one homozygous line with mutated *SMXL6/7/8* (T₃ generation) was selected and used for phenotyping assays. For *pBRC1wt::gBRC1* and *pBRC1m::gBRC1* constructs, more than ten independent transgenic lines of T₁ generation were used for phenotypic and gene expression assays.

All the primers used for plasmid construction are shown in Supplementary Table 1.

**RNA extraction and gene expression analysis.** For gene expression assay, tissues were frozen in liquid nitrogen and total RNA was extracted using the Trizol reagent (Invitrogen, USA). For total RNA extraction, axillary buds were harvested for *BRC1* and leaves from 3-week-old plants for *FHY3, SPL9,* or *SMXL6*. The cDNA was synthesized using the FastQuant RT Kit (Tiangen Biotech, China) and the expression levels were detected using the Transstart Tip Green qPCR Super Mix (TransGen Biotech, China) on the Applied Biosystems Q3 real-time PCR system according to the manufacturers' instruction. The *PP2A* gene (AT1G13320) was used as an internal control and the relative transcript levels were calculated using the comparative Ct method[51].

**Y1H, Y2H, BiFC, and LCI assays.** For Y1H assay, the *AD-SPL* and *2µ-pBRC1* or *2µ-pBRC2* plasmids were co-transformed into the yeast strain EGY48 as described in Lin et al[27]. For Y2H, the various plasmids for *LexA-SPLs* and *AD-FHY3* or *AD-SMXLs* were co-transformed into the yeast strain EGY48 plus H18 and grown on the SD/-Ura/-Trp/-His medium. The positive transformants were further selected on the medium containing raffinose, galactose, and 5-Bromo-4-chloro-3-indolyl-beta-D-galactopyranoside (Amresco, USA) as the substrate for blue color development.

For BiFC, the *A. tumefaciens* stain EHA105 harboring the *pSPYCE-SPLs* or *pSPYNE-FHY3, FAR1,* or *SMXLs* constructs were grown at 28 °C for 2–3 days. Then the positive transformants were cultured in liquid medium and the overnight cultures were diluted to OD₆₀₀ of 0.5 in a resuspension buffer containing 10 mM MgCl₂, 10 mM MES, and 100 mM acetosyringone and syringe-infiltrated into 4-week-old *N. benthamiana* leaves. The fluorescence within the infiltrated regions was visualized after 48 h using a Zeiss 510 Meta confocal laser scanning microscope.

For LCI, the *A. tumefaciens* stain EHA105 harboring *pCAMBIA1300-cLUC-SPLs* or *pCAMBIA1300-nLUC-FHY3, FAR1,* or *SMXLs* constructs were cultured in liquid medium at 28 °C for 2–3 days. After dilution with the resuspension buffer above, the cultures were syringe-injected into *N. benthamiana* leaves and incubated at 25 °C for 2–3 days. For the luciferase luminescence imaging, the infiltrated regions of the leaves were sprayed with 20 mg mL⁻¹ potassium luciferin (Gold Biotech, USA) and the luciferase luminescence was imaged using Night SHADE LB 985 system (Berthold, Germany).

**Luciferase activity assay.** The luciferase activity assay was performed following the method of Xie et al[24]. Briefly, *N. benthamiana* leaves were co-injected with *35S::SPLs-SPYCE* and *pBRC1::LUC* and incubated at 25 °C for 2–3 days. Then the injected leaves were detached and sprayed with 2 mg mL⁻¹ potassium luciferin (Gold Biotech, USA). The luciferase luminescence from the infiltrated area was imaged using Night SHADE LB 985 system (Berthold, Germany).

**Gel mobility shift assay.** The direct binding of SPL9 and SPL15 to the *BRC1* promoter was detected using an EMSA kit (Beyotime, China) following the manufacturer's protocol with probes listed in Supplementary Table. The GST and MBP proteins were used as the controls.

**Recombinant protein production.** For protein expression and purification, the SPL fragment containing SBP domain amplified from AD-SPLs above was cloned into the pGEX-4T-1 vector at the *EcoRI* restriction site to generate the *GST-SPL9SBP* and *GST-SPL15SBP* construct, respectively. The FHY3 fragment FHY3-I (containing the first 251 amino acids) and the full-length *SMXL6* was cloned into the pET28a vector to generate the *6×His-FHY3-I* and *6×His-SMXL6* constructs. All the constructs were transformed into the *Escherichia coli* strain Transette (TransGen Biotech, China). For recombinant protein expression, the bacteria strains were incubated in Luria-Bertain (LB) medium at 37 °C overnight and diluted at 1:100 with fresh LB medium and incubated at 37 °C for about 3 h until an optical density of 600 nm of 0.6–0.8 was reached. These bacteria were kept in the ice to cool down. Then the recombinant proteins GST-SPL9SBP, GST-SPL15SBP, or 6×His-FHY3-I were induced by 0.4 mM isopropyl β-D-thiogalactopyranoside (IPTG) at 16 °C overnight with gentle rotation (150 rpm), while the fusion proteins 6×His-SMXL6 were induced by 0.1 mM IPTG at 4 °C for more than 20 h with gentle shaking (80 rpm). The cells were harvested by centrifugation at 3214 × g for 10 min at 4 °C. Then the pellets were resuspended in PBS buffer (137 mM NaCl, 2.7 mM KCl, 10 mM Na₂HPO₄, 1.76 mM KH₂PO4, pH 7.4) for GST tag proteins or HIS Binding Buffer (250 mM KCl, 5 mM imidazole, 25 mM Tris-Cl, pH 8.0) for His tag proteins, respectively. After being ultrasonicated, the supernatant of cell debris was recovered by centrifugation at 13,000 g for 60 min. For GST tagged recombinant proteins, the supernatant was applied onto glutathione sepharose resin (GE Healthcare, USA) according to the manufacturer's protocol. After extensive washing with PBS buffer, the GST fusion proteins were eluted with GST Elution Buffer (20 mM reduced glutathione, 100 mM Tris-Cl, pH 8.0). For His tagged fusion proteins, the supernatant was applied onto Ni-NTA His-Bind resin (Novagen, USA) following the manufacturers' instruction. After extensive washing with HIS Binding Buffer, the His tagged fusion proteins were eluted with HIS Elution Buffer (250 mM KCl, 200 mM imidazole, 25 mM Tris-Cl, pH 8.0). These eluted proteins were then dialyzed against dialysis buffer (5 mM Tris-Cl, pH 8.0, 50 mM KCl), concentrated by ultrafiltration, aliquoted, frozen in liquid nitrogen and stored at −80 °C. The LC/MS/MS and Q-TOF analysis and the western blot of these proteins were provided in Supplementary Figs. 22 and 23 and Supplementary data set.

**ChIP-qPCR assay.** Eight-day-old transgenic seedlings of *35S::SPL9-HA* or *35S::SPL15-HA* were cross-linked with 1% formaldehyde and ground in liquid nitrogen. The chromatin complex was prepared following the method of Xie et al[24]. In brief, the supernatant was pre-cleared with 40 µl Protein-A-Agarose (16-157, EMD Millipore Corp.) and incubated at 4 °C for 1 h. Then the supernatant was moved into a microtube and 5 µl HA tag monoclonal antibodies (CB100005M, Cali-Bio) were added. After incubation at 4 °C for overnight with gentle agitation, 40 µl Protein-A-Agarose was added and incubated for 2 h. After washing, the immuno complex was eluted from the agarose beads. The precipitated DNA was then recovered and quantified using quantitative PCR with their individual primer pairs. The values were standardized to the input DNA to obtain the enrichment fold. *PP2A* was used as the internal control.

**Western blotting analysis.** For western blotting, 8-day-old seedlings were ground in liquid nitrogen and homogenized in extraction buffer (50 mM Tris-Cl, pH 7.4, 100 mM NaCl, 10% glycerol, 0.1% Tween-20, 1 mM DTT, 1 mM phenylmethylsulphonyl fluoride, and complete protease inhibitor cocktail). After spinning, the supernatant was collected and the total protein quantified using the Bradford method. The purified anti-FLAG antibodies (M185-7, MBL, Japan) with 1:5000 dilution were used to detect FHY3 or FAR1 proteins. For the actin internal control, the plant actin monoclonal antibodies (BE7008, EASYBIO, China) at a dilution of 1:5000 were used. The western blotting was imaged with the Celvin S 420 Chemiluminescence Imaging System (Biostep, German).

**Pull-down and Co-IP assay.** For pull-down assay, the purified GST-SPL9SBP or GST-SPL15SBP fusion protein was added to the total protein extract of His-FHY3-I or His-SMXL6, and then incubated for 1 h at 4 °C under rotation. After washing with the protein extract buffer for three times, the samples were boiled with loading buffer and run on 10% SDS-PAGE gels for separation. The anti-GST antibodies (PM013-7, MBL, Japan) and anti-His antibodies (D291-7, MBL, Japan) with 1:5000 dilution were used to detect the protein level of SPL-SBP and FHY3-I/SMXL6, respectively.

For Co-IP, *N. benthamiana* leaves infiltrated with *Agrobacterium* harboring the constructs *SPL9/15-MYC* and *FHY3/SMXL6-HA* were collected and homogenized in 2 mL of co-IP buffer (50 mM Tris-Cl, pH 7.5, 100 mM NaCl, 2 mM DTT, 0.1% Tween-20, 1 mM PMSF, and complete protease inhibitor cocktail). After centrifugation at 12,000 g for two times (10 min each time), the supernatant was collected and incubated for 2 h at 4 °C with anti-HA magnetic agarose beads (M180-10, MBL, Japan). After washing for three times with 1 mL of Co-IP buffer, the beads were boiled with loading buffer and run on 10% SDS-PAGE gels for separation. For western blotting, anti-MYC antibodies (M047-7, MBL, Japan), and anti-HA antibodies (561-7, MBL, Japan) with 1:5000 dilution were used to detect the levels of SPL9/15 or FHY3/SMXL6 proteins, respectively.

**Measurement of relative fluorescence**. The average fluorescence pixel intensity of each root was determined using Image J software (http://rsbweb.nih.gov/ij/; version 1.38) as described by Ye et al.[52,53]. In brief, draw a close freehand line using the "straight" tool in the software along the outline of individual root and then do "subtract background" and calculate the average fluorescence using the "measure" tool.

**Statistical analysis**. All real-time PCR and other quantitative analysis were repeated at least three times. To evaluate the significant difference among genotypes treated with or without EOD-FR, the method of two-way analysis of variance with interaction was adopted by using SPSS software (version IBM SPSS Statistics 22.0). The least significance difference (LSD) test was adopted to analyze the significant differences among multiple groups. The Student $t$ test was used to analyze the significant difference between two groups. The graphs with black dots and bars were produced using GraphPad Prism software (version 7.04) by entering the raw data.

## Data availability

All data supporting the findings of this study are included within the article and its Supplementary Information files or available upon request from the authors. The source data for Figs. 1b–e, 2b–e, 3c–d, 4b–d, f, 5b–e, 6b, e and Supplementary Figs. 1–5, 7–9, 11 and 12, 14, 16, 18, 23 are provided in the Source Data file.

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

## Acknowledgements
This work was supported by grants from National Natural Science Foundation of China (31770210 and 31570191) and National Key Research and Development Program of China (2016YFD0100303).

## Author contributions
H.Wang and Y.X. designed the research; Y.X., Y.L., M.M., Q.Z., Y.Z., B.Z., B.W. and H.Wei performed the research and analyzed the data; H.Wang and Y.X. wrote the paper.

## Competing interests
The authors declare no competing interests.
