## [Peer Review File · Nature Communications]

Reviewers' comments:

Reviewer #1 (Remarks to the Author):

Xie et al. dissected the involvement of FHY3 and FAR1 in the regulation of branching in response to light quality change. *fhy3/far1* has reduced branching under white light but diminished the difference in branching under white light versus EOD-FR shown for wild type. Inside to the story, FHY3 and FAR1 inhibit the binding of SPL9 and SPL15 to the BRC1 promoter, and SPL9 and SPL15 directly activates BRC1 expression. Meanwhile, three SMXL proteins also interact with SPL9 and SPL15 but rather repress their transactivation activity. Finally, FHY3 and FAR1 bind the first exons of SMXL6 and SMXL7 and activate their expression. A model is followed to present their overall discoveries.

Major comments

1) In Figure 3d, another control, MYC plus FHY3:HA, is needed for in vivo co-IP on the right panel. In addition, in vivo co-IP with FAR1 may show a low affinity toward either SPL9 or SPL15. The in vivo results with FAR1 may partially explain the subtle phenotype of *far1*. In Figure 6d, FAR1 appears to have a weak transactivation activity for SMXL6 and 7 compared to that of FHY3.

2) In Seven-day-old seedlings grown under WL conditions with EOD-FR treatment, both FHY3 and FAR1 proteins accumulated much less as shown in Figure 1d and 1e. Similar results were also shown for adult plants of four-week old in Figure S1. In Figure 4, EOD-FR may affect the ChIP of SPL9 and SPL15 to the BRC1 promoter. The authors may choose either wild type or FHY3 OX plants and variable ages to detect the best responses.

3) In Figure 6, the model suggests that "simulated shade treatment reduces the accumulation of FHY3 protein, leading to increased expression of BRC1 and reduced branching". As EOD-FR affects FHY3 and FAR1 accumulation, EOD-FR may affect the expression of SMXL6 and SMXL7 in either wild type or FHY3 OX. Data with EOD-FR treatment added in this figure plus experiments suggested in 2) would strengthen the model.

Minor points

1) In line 705, revise as both FHY3 and FAR1 protein level rapidly declined in seedlings treated with EOD-FR.

2) In line 454, fill the missing parts for "2_-pBRC1 or 2_-pBRC2".

Reviewer #2 (Remarks to the Author):

This manuscript describes some potentially very interesting results regarding the integration of light signalling into the shoot branching pathway in *Arabidopsis*. The authors propose that FHY3 and FAR1 regulate branching both by blocking SPL9:SPL15 regulation of BRC1, and by promoting transcription of SMXL6/7, which also block SPL9:SPL15 regulation of BRC1. Along the way, this involves showing that SPL9/SPL15 regulate BRC1 in *Arabidopsis* (not shown before, but homologous to rice), and that SMXL6/7 act through SPL9/SPL15 (again homologous to rice, not previously shown in *Arabidopsis*). So there is potentially a lot of interesting work here

Each part of the manuscript uses multiple different approaches to support its conclusions, and superficially, the manuscript seems very convincing. However, the manuscript lacks a substantial amount of methodological detail regarding what was actually done, and how it was done. I have major concerns about some of the results, because although it is clear what the result shows it is not clear how the results were obtained.

In particular, the authors claim to have expressed full length SMXL6 protein using a simple bacterial expression system. This is widely acknowledged to be impossible among labs working on strigolactone signalling. I simply don't believe that the authors expressed the protein as described in the manuscript. The SMXL6 protein doesn't appear to be the correct size

either - I'd expect a full-length His-SMXL6 from pET28a to be about 111 kDa (107.8 kDa for SMXL6 plus 3.4 kDa extra for the His-tag and linker residues), but the band here is under 100 kDa. It also isn't particularly pure (none of the "pure" proteins are), and the coomassie gel looks to me like it might have been altered to make the contaminant bands less prominent compared to the desired bands.

To credit the authors purification work, I would want to see:

- 1) the exact mass and sequence of the SMXL6 fusion protein they express
- 2) more information on their expression methods ie. growth medium, extraction method, extraction buffer (plus any additives)
- 3) raw Coomassie gels of their expression with ladders, including induced vs uninduced cells, and soluble vs insoluble fractions for both, plus any westerns they might have done (eg anti-His) with ladders
- 4) raw Coomassie gels of their purification with ladders, including crude, flow through, wash and elution fractions, plus any westerns they might have done (eg anti-His) with ladders
- 5) confirmation whether they did any secondary purifications prior to that gel, and any related evidence

The same applies to some extent for the other protein work: the supporting evidence and methods are simply not up to scratch.

I would also highlight that the genetic crosses the authors perform to support their protein work are largely uninformative, and do not provide any evidence for genetic interactions.

1) BRC1 vs SPL9/15

The *brc1 brc1 x rSPL9* OE is additive, and *BRC1OE spl9 spl15* is epistatic - no evidence for interaction here. Indeed, the evidence is more in favour of non-interaction.

2) FHY3/FAR1 vs SPL9/SPL15

Again, all the crosses of *fhy3 far* and *FHY3OE x spl9 spl15* demonstrate is that *spl9 spl15* is epistatic to *FHY3*. That in itself isn't good evidence for an interaction.

3) SMXL678 vs SPL9/SPL15

Again, the results just show that *SPL9/SPL15* are epistatic to *SMXL678*. The authors state that "while *SMXL6D-OE/ SPL9-OE* had more rosette branches than *SPL9-OE*", but that is absolutely not what the data in Fig 6 shows.

Our point-to-point responses to the reviewers' comments are listed below:

Reviewer #1 Remarks to the Author

Major comments:

Q1) *In Figure 3d, another control, MYC plus FHY3:HA, is needed for in vivo co-IP on the right panel. In addition, in vivo co-IP with FAR1 may show a low affinity toward either SPL9 or SPL15. The in vivo results with FAR1 may partially explain the subtle phenotype of far1. In Figure 6d, FAR1 appears to have a weak transactivation activity for SMXL6 and 7 compared to that of FHY3.*

Response: Thanks for helpful suggestions. As suggested, we re-performed the Co-IP assay with the new MYC control plus FHY3-HA (See new Fig. 3d). We also performed the Co-IP assay for the *in vivo* interaction between FAR1 and SPL9/SPL15 and provided the data in the revised manuscript (See new Fig. 3d). In Figure 6d, we also quantified the luciferase activity by the LUC/REN assay with internal reference (35S::REN) (See Supplementary Fig. 18), which shows that indeed FAR1 has a weaker transactivation activity for SMXL6 and SMXL7 than FHY3.

Q2) *In Seven-day-old seedlings grown under WL conditions with EOD-FR treatment, both FHY3 and FAR1 proteins accumulated much less as shown in Figure 1d and 1e. Similar results were also shown for adult plants of four-week old in Figure S1. In Figure 4, EOD-FR may affect the ChIP of SPL9 and SPL15 to the BRC1 promoter. The authors may choose either wild type or FHY3 OX plants and variable ages to detect the best responses.*

Response: Thanks for the valuable suggestions. As suggested, we conducted the ChIP-qPCR assay using seven-day-old and four-week-old WT, FHY3-OE and *fhy3 far1* double mutants carrying the 35S:*rSPL9-HA* transgene grown under normal white light and simulated shade conditions. We found that the *BRC1* DNA fragment was

more enriched in four-week-old adult plants grown under EOD-FR conditions by SPL9, indicating that indeed SPL9 can more effectively bind to the *BRC1* promoter under EOD-FR conditions to promote *BRC1* expression (See new Fig. 4c and Supplementary Fig. 11).

Q3) *In Figure 6, the model suggests that “simulated shade treatment reduces the accumulation of FHY3 protein, leading to increased expression of BRC1 and reduced branching”. As EOD-FR affects FHY3 and FAR1 accumulation, EOD-FR may affect the expression of SMXL6 and SMXL7 in either wild type or FHY3 OX. Data with EOD-FR treatment added in this figure plus experiments suggested in 2) would strengthen the model.*

Response: Thanks for helpful suggestions. As suggested, we compared the transcript levels of *SMXL6* and *SMXL7* between wild type, *fhy3 far1*, and *FHY3-OE* plants grown under normal white light and simulated shade conditions. As expected, we found that the expression levels of *SMXL6* and *SMXL7* significantly declined in wild type and *FHY3-OE* plants grown under simulated shade conditions compared with their counterparts grown under normal white light conditions (**Supplementary Fig. 6e**). These results further strengthened our model.

Minor points

Q1) *In line 705, revise as both FHY3 and FAR1 protein level rapidly declined in seedlings treated with EOD-FR.*

Response: Thanks. We corrected the sentence in Line 705 as “the FHY3 protein level rapidly declined in seedlings treated with EOD-FR”.

Q2) *In line 454, fill the missing parts for “2_-pBRC1 or 2_-pBRC2”.*

Response: Thanks for careful reading. We corrected them as “2 μ -pBRC1 or 2 μ -pBRC2” in Line 454.

Reviewer #2 (Remarks to the Author):

-----However, the manuscript lacks a substantial amount of methodological detail regarding what was actually done, and how it was done. I have major concerns about some of the results, because although it is clear what the result shows it is not clear how the results were obtained. In particular, the authors claim to have expressed full length SMXL6 protein using a simple bacterial expression system. This is widely acknowledged to be impossible among labs working on strigolactone signalling. I simply don't believe that the authors expressed the protein as described in the manuscript. The SMXL6 protein doesn't appear to be the correct size either - I'd expect a full-length His-SMXL6 from pET28a to be about 111 kDa (107.8 kDa for SMXL6 plus 3.4 kDa extra for the His-tag and linker residues), but the band here is under 100 kDa. It also isn't particularly pure (none of the "pure" proteins are), and the coomassie gel looks to me like it might have been altered to make the contaminant bands less prominent compared to the desired bands. To credit the authors purification work, I would want to see:

Q1) the exact mass and sequence of the SMXL6 fusion protein they express

Response: We understand the reviewer's concern and indeed we also found that SMXL6 protein is hard to be induced and expressed in *E. coli*. We apologize for not having provided the necessary details of protein expression and purification. We now provided detailed the information about our protein expression and purification procedure and conditions in the Materials and Methods section of the revised manuscript, including the growth medium, induction and extraction procedure of proteins, and components of each buffer used for protein purification (See Page 18, Line 500-507; Page 19, Line 510-511). We also performed LC-MS/MS and Q-TOF assays to confirm the identities of SMXL6 and other recombinant proteins as requested, including GST-SPL9SBP, GST-SPL15SBP and 6xHis-FHY3-I (**Supplementary Fig. 23; Supplementary data set**). We re-purified all proteins used in this study, including GST, MBP, GST-SPL9SBP, GST-SPL15SBP, 6xHis-FHY3-I and 6xHis-SMXL6, and re-ran their coomassie gels and conducted western blotting as

requested (**Supplementary Fig. 22**). All these data verified the identities of the purified target proteins.

As a note, we firstly used *Escherichia coli* both BL21 (DE3) and Tuner stains to induce SMXL6 protein expression with different incubation temperatures (16 and 37 °C) and different working concentrations of isopropyl β-D-thiogalactoside (IPTG) (0.1, 0.5, 1 and 5 mM), however, no obvious expression of SMXL6 proteins was detected under these conditions. Then we turned to the *E. coli* Transette stain (modified from Rosette stain) to induce SMXL6 protein expression with various concentrations of IPTG at 16 and 37 °C, respectively. We found that expressed SMXL6 protein only existed in pellet but not in the supernatant. Finally, we set 4 °C as the induction temperature (the actual temperature in shaker is around 5-6 °C) and used 0.1 mM IPTG to induce SMXL6 protein expression. We incubated the bacteria for more than 20 h with gentle rotation (80 rpm) and then harvested the cells. After sonication and purification, we obtained recombinant SMXL6 proteins used for all the assays (coomassie staining, western blot analysis, and LC-MS/MS and Q-TOF assays).

Q2) *more information on their expression methods ie. growth medium, extraction method, extraction buffer (plus any additives)*

Response: See reply to Q1.

Q3) *raw Coomassie gels of their expression with ladders, including induced vs uninduced cells, and soluble vs insoluble fractions for both, plus any westerns they might have done (eg anti-His) with ladders*

Response: Thanks for helpful suggestions. As suggested, we re-purified the protein and re-ran raw coomassie gels including crude, flow through, wash and elution proteins and conducted western blotting. We supplied the data in **Supplementary Fig. 22**.

Q4) *raw Coomassie gels of their purification with ladders, including crude, flow*

through, wash and elution fractions, plus any westerns they might have done (eg anti-His) with ladders

Response: See our reply to Q3.

Q5) *confirmation whether they did any secondary purifications prior to that gel, and any related evidence*

Response: Thanks. We did not perform any secondary purifications in this study.

The same applies to some extent for the other protein work: the supporting evidence and methods are simply not up to scratch.

I would also highlight that the genetic crosses the authors perform to support their protein work are largely uninformative, and do not provide any evidence for genetic interactions.

Q1) *BRC1 vs SPL9/15*

The brc1 brc1 x rSPL9 OE is additive, and BRC1OE spl9 spl15 is epistatic - no evidence for interaction here. Indeed, the evidence is more in favour of non-interaction.

Response: To verify the genetic relationship between SPL9/15 and BRC1, we grew *brc1 brc2*, *spl9 spl15*, *BRC1-OE*, *rSPL9-OE*, and their high order mutants in large scale (more than 50 plants for each genotype) and counted their rosette-leaf branch number. The results showed that that there was no significant difference in the rosette-leaf branch number between *brc1 brc2/ rSPL9-OE* and *brc1 brc2*, and no significant difference between *BRC1-OE/ spl9 spl15* and *BRC1-OE* (**new Supplementary Fig. 4**). In addition, we performed RT-qPCR assay, which showed that the *BRC1* expression levels in *spl9 spl15* and *brc1 brc2/ rSPL9-OE* plants were significantly lower than that in wild type, while the *BRC1* expression levels in *rSPL9-OE* and *BRC1-OE/ spl9 spl15* plants were significantly higher than that in wild type (**Supplementary Fig. 5**). These data strongly suggest that *SPL9/15* act upstream of *BRC1* to inhibit branching. To further confirm this, as the editor suggested, we generated the constructs containing the *BRC1* coding regions driven by the wild type

BRC1 promoter (*pBRC1wt::gBRC1*) or *BRC1* promoter with the SBP binding site mutated (*pBRC1m::gBRC1*) and transformed these constructs into *brc1 brc2*, *spl9 spl15/ brc1 brc2*, *rSPL9-OE/ brc1 brc2* backgrounds and examined their effects on branch numbers. We found that the branch number in the *brc1 brc2* plants harboring the *pBRC1wt::gBRC1* transgene was restored to wild type level, whereas the *brc1 brc2* plants harboring the *pBRC1m::gBRC1* transgene had significantly more branches than the wild type plants (similar to *brc1 brc2*). We also found that the *spl9 spl15/ brc1 brc2* plants carrying the *pBRC1wt::gBRC1* or *pBRC1m::gBRC1* transgenes had similar branch number as the *spl9 spl15/ brc1 brc2* plants (**Supplementary Fig. 8**). Moreover, we found that *rSPL9-OE/ brc1 brc2* plants harboring *pBRC1wt::gBRC1* transgene had significantly fewer branches than *brc1 brc2* and wild type plants. However, no significant difference was detected between *rSPL9-OE/ brc1 brc2* plants harboring the *pBRC1m::gBRC1* transgene and the *brc1 brc2* mutant plants (**Supplementary Fig. 8**). Collectively, these results strongly support the notion that *SPL9* and *SPL15* act to regulate branch number through regulating *BRC1* expression via binding to the the SBP core motif in the *BRC1* promoter.

Q2) *FHY3/FAR1* vs *SPL9/SPL15*

Again, all the crosses of *fhy3 far1* and *FHY3OE x spl9 spl15* demonstrate is that *spl9 spl15* is epistatic to *FHY3*. That in itself isn't good evidence for an interaction.

Response: We believe that our multiple molecular assays (including yeast two-hybrid, BiFC, LCI, pull-down, Co-IP) provided substantial evidence to support the notion that *FHY3/FAR1* directly interact with *SPL9/15*. For genetic relationship assay, we found that both *fhy3 far1/spl9 spl15* and *FHY3-OE/spl9 spl15* plants had similar number of rosette branches as the *spl9 spl15* double mutant, suggesting that *FHY3* and *FAR1* act upstream of *SPL9* and *SPL15* to promote rosette branching. We also further examined the *BRC1* transcript levels in *fhy3 far1*, *spl9 spl15*, *FHY3-OE*, *rSPL9-OE*, and their high order mutant plants. The results showed that both *fhy3 far1/ spl9 spl15* and *FHY3-OE/spl9 spl15* had similar *BRC1* expression levels as *spl9 spl15* (**See**

Supplementary Fig. 12). To test a possible effect of FHY3/FAR1 on the transcription of *SPL9/15*, we conducted yeast one-hybrid assay to test the binding of FHY3/FAR1 to the promoters of *SPL9* and *SPL15*. The result showed that no binding between FHY3/FAR1 and the promoters of *SPL9/15* (See **Supplementary Fig. 19**). This observation is consistent with the observation that there is no typical FHY3/FAR1 binding site (FBS, 5'-CACGCGC-3') in the promoters of *SPL9/15*, suggesting that FHY3/FAR1 do not bind to the promoters of *SPL9/15*. Further RT-qPCR assay showed that there is no significant difference in the transcript levels of *SPL9* or *SPL15* between wild type, *fhy3far1* and *FHY3-OE* plants (See **Supplementary Fig. 12**), suggesting that FHY3/FAR1 do not transcriptionally regulate the expression of *SPL9/15*. Collectively, we believe our data support the notion that FHY3/FAR1 regulate *BRC1* expression through physically interacting with *SPL9/15*.

Q3) *SMXL678* vs *SPL9/SPL15*

*Again, the results just show that *SPL9/SPL15* are epistatic to *SMXL678*. The authors state that “while *SMXL6D-OE/ SPL9-OE* had more rosette branches than *SPL9-OE*”, but that is absolutely not what the data in Fig 6 shows.*

Response: We also believe that our multiple molecular assays provided substantial evidence to support the notion that *SMXL6/7* directly interact with *SPL9/15*. In addition, we found that the *smxl6/7/8/spl9/15* quintuple mutant had similar branch number as *spl9 spl15*, while *SMXL6D-OE/ SPL9-OE* had similar branch number as *SPL9-OE*, suggesting that *SPL9/15* act downstream of *SMXL6/7*. Moreover, RT-qPCR assay showed that *smxl6/7/8/spl9/15* had similar *BRC1* transcript level as *spl9/15*, while *SMXL6D-OE/ SPL9-OE* had similar *BRC1* transcript level as *SPL9-OE* (See **Supplementary Fig. 16**), supporting the notion that *SPL9* acts downstream of *SMXL6/7*. To test whether *SMXL6/7* regulate *SPL9/15* expression, we also conducted the yeast one-hybrid assay to test whether *SMXL6/7* bind to the promoters of *SPL9* and *SPL15*. The result showed that no binding between *SMXL6/7* and the promoters of *SPL9/15* was detected (See **Supplementary Fig. 19**), suggesting that *SMXL6/7* do not bind to the promoters of *SPL9/15* and transcriptionally regulate their expression.

Further, no significant difference was detected in the expression levels of *SPL9* or *SPL15* between wild type, *smxl6/7/8* and *SMXL6D-OE* plants (**See Supplementary Fig. 16**). Together, all results support the notion that *SMXL6/7* regulate *BRC1* expression through direct physical interaction with *SPL9/15*.

To test the possibility whether *FHY3/FAR1* and *SMXL6/7* directly regulate *BRC1* expression, we also conducted two sets of additional experiments. First, we tested whether *FHY3/FAR1* and *SMXL6/7* could directly bind to the *BRC1* promoter by bypassing *SPL9/15*. However, yeast one-hybrid assay revealed that there was no direct binding of *FHY3/FAR1* or *SMXL6/7* to the *BRC1* promoter (**See Supplementary Fig. 19**). This result suggests that it is unlikely *FHY3/FAR1* and *SMXL6/7* directly regulate *BRC1* transcription. Second, we tested whether *FHY3/FAR1* and *SMXL6/7* could regulate *BRC1* activity through protein-protein interaction. Our yeast two-hybrid assay showed that no direct interaction was detected between *BRC1* and *FHY3/FAR1* or *SMXL6/7* proteins (**See Supplementary Fig. 20**). These results are consistent with our proposition that both *FHY3/FAR1* and *SMXL6/7* do not directly regulate *BRC1* gene expression/activity.

We also noted that the *SMXL6D-OE/SPL9-OE* plants had slightly more (~15.7%) branches than *SPL9-OE* plants, suggesting that *SMXL6D* may affect branching number through additional targets (such as *SPL15*) besides *SPL9*. This observation is still consistent with our proposition that *SMXL6/7* act upstream of *SPL9/15* to regulate branching. To be more accurate, we modified the sentence “while *SMXL6D-OE/SPL9-OE* had more rosette branches than *SPL9-OE*” as “**while *SMXL6D-OE/SPL9-OE* had slightly more (~15.7%) rosette branches than *SPL9-OE***”.

REVIEWERS' COMMENTS:

Reviewer #1 (Remarks to the Author):

The authors have addressed my comments.

Reviewer #2 (Remarks to the Author):

In this revised manuscript, the authors have addressed, or claimed to have addressed, all my criticisms from my previous review.

However, I continue to have concerns about some of experiments and the incomplete documentation provided by the authors. Here is a sample of concerns:

1) The manuscript uses a huge number of different transgenic Arabidopsis lines. How were these transformed? What strategy did they use to make stable T3 homozygous lines? How many independent transgenic lines did they examine for each construct? What percentage showed the phenotype? Which line did they pick for the final analysis, and why?

Based on the speed at which they generated new constructs for this manuscript (e.g. pBRC1wt:gBRC1 and pBRC1m:gBRC1) -- less than 6 months -- the authors must have used T1 lines for this analysis. Thus each "sample" of a transgenic line would have actually consisted of separate, independent T1 transgenic lines, with different genomic insertions and therefore effects on gene expression. Some of the lines would have no phenotype at all. It is simply not appropriate to perform phenotypic experiments on a collection of independent T1 lines.

Is this also the case for all the other transgenic lines in the paper? Did the authors perform the experiments on a collection of freshly generated, independent T1 lines? Or did they use T3 lines? If I asked the authors for seed from the lines shown in the paper, would they actually be able to send me homozygous lines?

All of this is completely standard experimental practice when using transgenic Arabidopsis, and important methodological information to provide.

2) Statistics

The majority of the statistical methodology provided in the paper just says: "Different letters indicate significant differences by two-way ANOVA ($p < 0.01$). But ANOVA does not indicate where the differences occur, only that there are significant differences. So how did the authors assign these letters?

Did the authors check their data for normality and use the appropriate non-parametric test where needed? (a core requirement for publishing in Nature journals).

3) Figure manipulation

I would also draw the authors attention to Figure 5B, where in the top row, exactly the same leaf has apparently been infiltrated with SMXL7-cLUC AND SMXL8-cLUC, in exactly the same splotch pattern, but with different signal intensities. Of course, these are just the same image with different exposures, masquerading as separate experiments.

This is not a freak mistake, because on the lower row of Figure 5B, exactly the same leaf has apparently been infiltrated with SMXL6-cLUC AND SMXL7-cLUC, in exactly the same splotch pattern, but again with different signal intensities. Again, the same image with different exposures is being used to cover two experiments.

I would suggest that the authors need to take a very careful look at all their images, to make sure there are no more occurrences of image duplication between experiments. Obviously this is not acceptable.

Point-to-point response to reviewers' comments:

Reviewer #1:

The authors have addressed my comments.

Reply: We wish to thank reviewer #1 for his/her support towards this work.

Reviewer #2

Q1. The manuscript uses a huge number of different transgenic Arabidopsis lines. How were these transformed? What strategy did they use to make stable T3 homozygous lines? How many independent transgenic lines did they examine for each construct? What percentage showed the phenotype? Which line did they pick for the final analysis, and why? Based on the speed at which they generated new constructs for this manuscript (e.g. pBRC1wt:gBRC1 and pBRC1m:gBRC1) -- less than 6 months -- the authors must have used T1 lines for this analysis. Thus each "sample" of a transgenic line would have actually consisted of separate, independent T1 transgenic lines, with different genomic insertions and therefore effects on gene expression. Some of the lines would have no phenotype at all. It is simply not appropriate to perform phenotypic experiments on a collection of independent T1 lines. Is this also the case for all the other transgenic lines in the paper? Did the authors perform the experiments on a collection of freshly generated, independent T1 lines? Or did they use T3 lines? If I asked the authors for seed from the lines shown in the paper, would they actually be able to send me homozygous lines? All of this is completely standard experimental practice when using transgenic Arabidopsis, and important methodological information to provide.

Reply: Thanks for reminding us to clarify these issues. All the stable transgenic lines were generated using the floral dip method as described in Clough and Ben¹. Then the seeds of the transformed plants (designed as T₀) were harvested and sown on half strength Murashige and Skoog (1/2MS) media plates supplemented with 1% sucrose, 50 µg ml⁻¹ cefalexin (for inhibiting the growth of agrobacterium) and proper concentration of antibiotic suggested by the vendors. The survived seedlings (designated T₁ generation, typically we grew more than 30 plants for each construct, which are treated as independent lines) with green cotyledons and long root were transferred into soil to grow for T₁ seeds.

More than 100 T₁ seeds from individual T₁ line were sown on the 1/2 MS media plates containing antibiotic. After 7-10 days' growth, the seedlings (T₂) with yellow cotyledons and short or no roots (presumably transgene-negative control seedlings) were counted and the ratio of antibiotic-sensitive to antibiotic resistant seedlings was calculated. From the T₁ line with the segregation ratio ~1:3 (indicating a single transgene locus), more than 20 antibiotic resistant T₂ seedlings were transferred into soil to grow for T₂ seed harvest and their seeds were harvested separately from individual T₁ plant. The T₂ seeds from individual T₂ plant were again sown on the 1/2 MS media plates with antibiotic and that the lines did not show segregation again were deemed homozygous. Meanwhile, we also plated the T₂ seeds from individual T₂ plant on the 1/2 MS media plates without antibiotic as duplicates and the homozygous seedlings grown on 1/2 MS media plates without antibiotic were used for further assay (to avoid effect of antibiotics). T₃ seeds are harvested from selfed homozygous T₂ plants and used for phenotypic assays. For the *smxl6/7/8* CRISPR vector, we totally obtained 57 individual T₁ transgenic lines. After PCR detection with the special primer pairs, however, we found there was no obvious deletion for *SMXL6* gene in most of these lines, although there were obvious or complete deletion for both *SMXL7* and *SMXL8* in 12 individual line. We sequenced the PCR products amplified with *SMXL6* primer pair using these 12 lines as template and found that two lines (Line #2 and Line #16) contain mutations in the *SMXL6* gene (Supplementary Figure 15). These two T₂ lines were propagated to obtain the homozygous *smxl6/7/8* mutant from their T₃ generation and used for further study.

As the reviewer correctly pointed out, the percentage of transgenic plants with the phenotype vary due to many genetic (such as copy number or position of insertions) and epigenetic effects. For the *FHY3-OE* construct, we obtained 30 positive T₁ lines but most of them (19 lines) showed a weak phenotype. While for *rSPL9-OE*, we obtained 26 positive T₁ lines and 22 of them showed obvious phenotype. For the newly generated constructs, the number of their independent transgenic events in different backgrounds is less than 20 (13, 11, 10, 14, 12, 16 events for the constructs *brc1brc2/ pBRC1wt::gBRC1*, *brc1brc2/ pBRC1m::gBRC1*, *spl9spl15 brc1brc2/ pBRC1wt::gBRC1*, *spl9spl15 brc1brc2/ pBRC1m::gBRC1*, *rSPL9-OE brc1brc2/ pBRC1wt::gBRC1*, *rSPL9-OE brc1brc2/ pBRC1m::gBRC1*, respectively) and the majority of them showed the phenotype.

Based on their phenotype and expression level in T₁ heterozygous plants, we usually selected 3-5 independent overexpression lines for each construct to obtain homozygous lines. During the period we also checked their phenotype. We re-analyzed their phenotype and expression levels in their homozygous lines to make sure that the expression level is associated with the phenotype. Based on the phenotypic analysis and expression detection, the line with the highest expression level and obvious phenotype is selected for further study. Actually, we used 2-3 lines for each construct at the beginning to do the phenotype analysis.

To save time, we used the T₁ lines of the new generated constructs for analyzing the roles of *BRC1* wild type promoter and mutated promoter, a practice used in previously published studies². We are also willing to share our reported materials in this manuscript upon request.

Q2. Statistics

The majority of the statistical methodology provided in the paper just says: "Different letters indicate significant differences by two-way ANOVA (p<0.01). But ANOVA does not indicate where the differences occur, only that there are significant differences. So how did the authors assign these letters? Did the authors check their data for normality and use the appropriate non-parametric test where needed? (a core requirement for publishing in Nature journals).

Reply: Thanks for valuable comments. We added a paragraph about the statistical analysis in the Methods section (**See Page 23, Lines 645-653**). We used the SPSS software (version IBM SPSS Statistics 22.0) and the two-sided least significance difference (LSD) test for multiple comparisons. We provided the analysis result in the Source Data file together with their raw data. To show the difference between two group, we used letters not symbol * since the same symbol can not tell all the differences between groups. We assign and label these letters according to the analysis result. In brief, we assign the highest value as "a". The second highest value will be assigned as "b" if there are significant differences between the highest value and the second highest value, otherwise assigned as "a" and so forth.

Q3. Figure manipulation

I would also draw the authors attention to Figure 5B, where in the top row, exactly the same leaf has apparently been infiltrated with SMXL7-cLUC AND SMXL8-cLUC, in exactly the same

splotch pattern, but with different signal intensities. Of course, these are just the same image with different exposures, masquerading as separate experiments. This is not a freak mistake, because on the lower row of Figure 5B, exactly the same leaf has apparently been infiltrated with SMXL6-cLUC AND SMXL7-cLUC, in exactly the same splotch pattern, but again with different signal intensities. Again, the same image with different exposures is being used to cover two experiments.

Reply: Thanks for pointing this out. It is a careless mistake and we apologize for that (we noticed it ourselves during the checking process and corrected it when we sent a revised manuscript to the editor to answer his quires last time). As suggested by the editor, we also provided the biological replicates as many as possible for all the fluorescence images including the luciferase activity assay and luciferase complementation imaging assay (**See the Supplementary pdf file in the Source Data file**).

Reference:

1. Clough, S. J. & Bent, A. F. Floral dip: a simplified method for *Agrobacterium* mediated transformation of *Arabidopsis thaliana*. *Plant J.* **16**, 735–743 (1998).
2. Harmer, S. L. & Kay, S. A. Positive and negative factors confer phase-specific circadian regulation of transcription in *Arabidopsis*. *Plant Cell* **17**,1926–1940 (2005).